# BRCA2 abrogation triggers innate immune responses potentiated by treatment with PARP inhibitors

Timo Reisländer [1], Emilia Puig Lombardi[2,3,7], Florian J. Groelly[1,7], Ana Miar[4], Manuela Porru[5], Serena Di Vito[5], Benjamin Wright[6], Helen Lockstone[6], Annamaria Biroccio[5], Adrian Harris [4], Arturo Londoño-Vallejo [2,3] & Madalena Tarsounas[1]

Heterozygous germline mutations in *BRCA2* predispose to breast and ovarian cancer. Contrary to non-cancerous cells, where *BRCA2* deletion causes cell cycle arrest or cell death, tumors carrying *BRCA2* inactivation continue to proliferate. Here we set out to investigate adaptation to loss of BRCA2 focusing on genome-wide transcriptome alterations. Human cells in which BRCA2 expression is inhibited for 4 or 28 days are subjected to RNA-seq analyses revealing a biphasic response to *BRCA2* abrogation. The early, acute response consists of downregulation of genes involved in cell cycle progression, DNA replication and repair and is associated with cell cycle arrest in G1. Surprisingly, the late, chronic response consists predominantly of upregulation of interferon-stimulated genes (ISGs). Activation of the cGAS-STING-STAT pathway detected in these cells further substantiates the concept that *BRCA2* abrogation triggers cell-intrinsic immune signaling. Importantly, we find that treatment with PARP inhibitors stimulates the interferon response in cells and tumors lacking BRCA2.

[1] Genome Stability and Tumourigenesis Group, The CR-UK/MRC Oxford Institute for Radiation Oncology, Department of Oncology, University of Oxford, Old Road Campus Research Building, Oxford OX3 7DQ, UK. [2] Institut Curie, PSL Research University, CNRS, UMR3244, F-75005 Paris, France. [3] Sorbonne Universités, UPMC Univ Paris 06, CNRS, UMR3244, F-75005 Paris, France. [4] Hypoxia and Angiogenesis Group, Weatherall Institute of Molecular Medicine, Department of Oncology, University of Oxford, Oxford OX3 9DS, UK. [5] Area of Translational Research, IRCCS Regina Elena National Cancer Institute, 00144 Rome, Italy. [6] Bioinformatics and Statistical Genetics Core, Wellcome Trust Centre for Human Genetics, University of Oxford, Oxford OX3 7BN, UK. [7] These authors contributed equally: Emilia Puig Lombardi, Florian J. Groelly. Correspondence and requests for materials should be addressed to M.T. (email: madalena.tarsounas@oncology.ox.ac.uk)

BRCA2 tumor suppressor plays key roles in cell physiology by promoting DNA replication and DNA double-strand breaks (DSBs) repair via homologous recombination[1]. The latter is well-characterized biochemically and relies on BRCA2 loading the RAD51 recombinase at sites of DSBs that have been processed by resection. Assembly of the RAD51 nucleoprotein filament facilitates the search and subsequent invasion of homologous DNA, which acts as a template for the repair reaction[2]. While BRCA2 function in DNA repair has been studied in the context of DNA damage inflicted by exposure to genotoxic agents (e.g., ionizing radiation, DNA cross linking agents), the role of BRCA2 in replication is intrinsic to cell physiology. In the absence of external challenges, loss of BRCA2 triggers a significant decrease in replication fork progression and a high frequency of stalled forks[3,4]. A subset of these forks collapse and are converted to DSBs which, due to compromised HR repair, accumulate in BRCA2-deficient cells. Moreover, nucleolytic degradation of the stalled forks can occur upon extensive replicative arrest[5,6]. Conceivably, this acute perturbation of DNA replication triggered by inactivation of BRCA2 could lead to rampant genomic instability, which is lethal or severely obstructs cell proliferation in primary human cells[7]. Likewise, disruption of *Brca2* gene in mice is embryonically lethal[8–10]. Mechanisms of replication stress and DNA damage tolerance mediate cellular adaptation to chronic loss of BRCA2, enable cells to survive and ultimately underlie their tumorigenic potential. Consistent with this, loss of *BRCA2* occurs in tumors and is thought to promote tumorigenesis, while *BRCA2* heterozygous germline mutations increase susceptibility to breast and ovarian cancer, as well as other cancers[11–13].

Here we investigate the possibility that transcriptional alterations provide modalities of cell adaptation to loss of BRCA2, thus preventing cell death or proliferative arrest. We characterized the transcriptome of BRCA2-deficient cells, using a doxycycline (DOX)-inducible shRNA to inhibit BRCA2 expression in human non-small cell lung carcinoma H1299 cells and invasive ductal breast cancer MDA-MB-231 cells. RNA-sequencing (RNA-seq) analyses conducted after 4 and 28 days of DOX-induced BRCA2 depletion enabled us to monitor the dynamics of gene expression and to identify substantial transcriptional alterations from early to late stages of BRCA2 inactivation. In the short term, we observe downregulation of cell cycle, DNA replication and repair genes, which correlates with marked accumulation of BRCA2-deficient cells in G1. In the long-term, we find that cell cycle re-entry occurs concomitantly with ISG upregulation. These are genes involved in the innate immune response and controlled by interferon signaling[14]. An ISG subset is upregulated in *BRCA2*-deleted primary ovarian tumors. These results support the concept that inducible BRCA2 inactivation in cultured cells recapitulates cellular changes associated with loss of BRCA2 during tumorigenesis and could provide clues to mechanisms of cellular adaptation in tumors or tumor vulnerabilities. Importantly, treatment with PARP inhibitors (olaparib, talazoparib) stimulates upregulation of the interferon signaling genes in vitro and in vivo.

## Results

**BRCA2 inactivation elicits changes in gene expression.** Previous studies have implicated BRCA2 in gene transcription via interaction with EMSY[15] and have reported differential gene expression between *BRCA2*-deleted and wild-type cells using expression microarrays[16]. *BRCA2*-deleted cells are terminally adapted and this precluded analysis of progressive changes associated with BRCA2 inactivation. Here we used RNA-seq analyses of gene expression to determine how the transcriptome of BRCA2-deficient cells is modulated over time. We cultured H1299 and MDA-MB-231 human cells carrying a DOX-inducible BRCA2 shRNA cassette in the presence or absence of 2 µg/mL DOX for 28 days and collected samples for RNA-seq analysis after 4 and 28 days of treatment (Fig. 1a). BRCA2 expression was effectively suppressed by DOX exposure in the short term (4 days), as well as in the long-term (28 days), as indicated by immunoblotting of cell extracts prepared at these time points (Fig. 1b).

Before proceeding with RNA-seq analyses of BRCA2-deficient cells, we performed control experiments in which we addressed whether DOX alone could induce transcriptional changes under our experimental conditions. Parental H1299 and MDA-MB-231 cells were incubated with 2 µg/mL DOX for 4 and 28 days, before collecting samples for RNA-seq analyses (Supplementary Fig. 1a, b). Hierarchical clustering of sample-to-sample Euclidean distances based on the RNA-seq data showed no clear clustering of samples treated with DOX ($n = 3$ independent experiments) or untreated samples ($n = 3$ independent experiments) in either cell line, indicative of insignificant alterations between the −DOX and

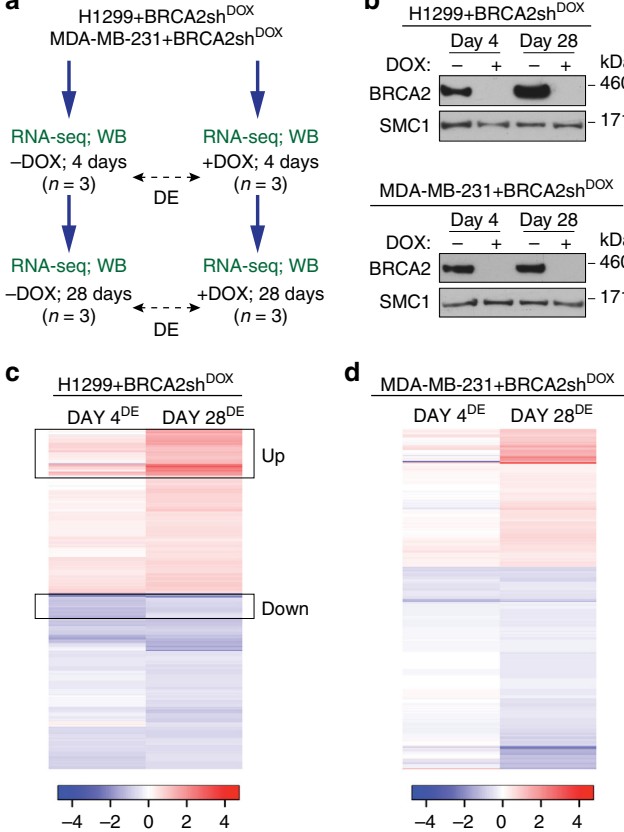

**Fig. 1** Transcriptome changes during the transition from short-term (acute) to long-term (chronic) *BRCA2* inactivation in human cells. **a** Human H1299 and MDA-MB-231 cells carrying a doxycycline (DOX)-inducible BRCA2 shRNA were grown in the presence or absence of 2 µg/mL DOX for 4 or 28 days before processing for RNA-seq and western blot analyses. **b** Whole-cell extracts prepared after 4 or 28 days of DOX treatment were immunoblotted as indicated. SMC1 was used as a loading control. **c**, **d** RNA-seq analyses of cells treated as in (**a**) identify transcriptional alterations specific to BRCA2-deficiency (FDR < 0.05) after 4 and 28 days of DOX treatment. Heatmaps depict Log$_2$(Fold Change) of top 480 genes differentially expressed in +DOX versus −DOX cells, after 4 or 28 days of DOX treatment. Boxes indicate a subset of genes downregulated at day 4 (DOWN) or upregulated at day 28 (UP) in BRCA2-deficient H1299 cells. $n = 3$ independent experiments; DE, differential expression

+DOX samples. Moreover, differential gene expression analyses with false discovery rate (FDR) of 5% revealed that only two genes were differentially expressed in DOX-treated versus untreated H1299 cells at day 4 and none at day 28 of DOX treatment (Supplementary Fig. 1c). The corresponding numbers of deregulated genes in MDA-MB-231 were 3 and 7, respectively (Supplementary Fig. 1d). These results demonstrate that DOX treatment for 4 or 28 days has a negligible impact on gene expression of parental cells lacking BRCA2 shRNA.

In contrast to this, DOX treatment inflicted substantial changes on the transcriptome of H1299 and MDA-MB-231 cells carrying a DOX-inducible BRCA2 shRNA. Hierarchical clustering of sample-to-sample Euclidean distances showed a clear distinction between +DOX ($n = 3$ independent experiments) and −DOX samples ($n = 3$ independent experiments) in both cell lines (Supplementary Fig. 2a, b), indicative of specific, substantial differences in the transcriptome upon BRCA2 abrogation. Consistent with this, DOX-inducible BRCA2 depletion in H1299 cells triggered significant alterations (FDR < 0.05, no fold-change filter) in the expression of 1363 genes at day 4 and of 479 genes at day 28 of DOX treatment (Supplementary Fig. 2c and Supplementary Data 1). The corresponding numbers of deregulated genes in MDA-MB-231 cells were 187 and 480, respectively (Supplementary Fig. 2d and Supplementary Data 2), suggesting that BRCA2 abrogation had a less pronounced effect on gene expression in the short term (4 days) in this cell line.

Next we conducted supervised clustering analyses of the top 480 genes differentially expressed (FDR < 0.05, no fold-change filter) in H1299 and MDA-MB-231 cells upon DOX-induced BRCA2 inactivation (Fig. 1c, d). We observed a clear variation in the differential transcription profiles between day 4 and day 28 of DOX treatment in both cell lines, which suggests that the transcriptome of BRCA2-deficient cells undergoes dynamic changes with time. The heatmaps showing the expression profiles of H1299 cells at replicate level are shown in Supplementary Fig. 3.

**Pathways deregulated upon short- versus long-term BRCA2 loss.** We proceeded to identify which gene sets are differentially expressed in BRCA2-deficient versus -proficient cells, at 4 and 28 days after shRNA induction. We focused on H1299 cells because DOX treatment had a high impact on gene expression in this cell line. We used stringent conditions (FDR < 0.05; |Log$_2$(Fold Change)| > 0.5) which enabled us to identify deregulated genes with high confidence (Fig. 2a, b). At day 4 of DOX treatment, this analysis identified 574 genes (42% of deregulated genes) significantly downregulated (Log$_2$(Fold Change) < −0.5; Supplementary Data 1) and 147 genes (11% of deregulated genes) significantly upregulated (Log$_2$(Fold Change) > 0.5) in BRCA2-deficient cells. At day 28, there were 194 genes (40%) significantly downregulated (Log$_2$(Fold Change) < −0.5; Supplementary Data 2) and 213 genes (44%) significantly upregulated (Log$_2$(Fold Change) > 0.5) in BRCA2-deficient H1299 cells.

Gene set enrichment analysis of differentially expressed genes based on functional annotation (Gene Ontology—Biological Process database) showed enrichment in specific pathways (Fig. 2c, d). Genes downregulated in BRCA2-deficient cells at day 4 were mainly implicated in cell cycle, chromosome segregation, DNA repair, and DNA replication, and defined an early, acute response to BRCA2 inactivation. The genes upregulated at day 28 primarily mediated cytokine and immune responses. Interestingly, the proliferation capacity of BRCA2-deficient cells (H1299 or MDA-MB-231) was not substantially reduced compared with their wild-type counterparts, as determined in population doubling assays (Supplementary Fig. 4).

To complement pathway mapping, we performed network analyses, which showed how the 574 downregulated genes interact with each other and cluster into different pathways (Supplementary Fig. 5). We retrieved high-confidence, experimentally validated protein–protein interactions through the NetworkAnalyst platform[17]. The network was generated by mapping the significant genes to the STRING interactome[18] and applying a search algorithm to identify first-order neighbors (proteins that directly interact with a given protein) for each of the mapped genes. We generated a highly connected first-order network consisting of 669 nodes and 2970 edges, with larger nodes indicating higher connectivity. For example, prominent DNA repair nodes identified key DNA repair factors, including RAD51, RAD52, BML, MUS81, FANCA, and MLH1, while main DNA replication nodes identified factors such as PCNA, MCM2, and RRM2. This indicates concerted action of the genes downregulated at day 4 in key cellular processes, such as control of cell cycle progression and genome integrity. Similar mapping of the 213 genes upregulated at day 28 generated a network with fewer interconnected nodes (446 nodes) and 1190 edges (Supplementary Fig. 5b). Pathway analysis after addition of direct interactors amplified enrichment in immune system related pathways (hypergeometric test $p$-value < 1e$^{-21}$). The top five impacted pathways were innate immune response (77 hits, $p$-value = 3.93e$^{-27}$), immune response (116 hits, $p$-value = 1.83e$^{-25}$), interaction with host (61 hits, $p$-value = 1.86e$^{-25}$), cytokine-mediated signaling (54 hits, $p$-value = 9.34e$^{-23}$), and defense response (112 hits, $p$-value = 2.9e$^{-21}$).

Following up on the RNA-seq results showing that cell cycle genes are downregulated after 4 days of DOX treatment, we evaluated the impact of BRCA2 loss on cell-cycle progression. FACS analyses of EdU-pulsed cells revealed pronounced alterations in the cell cycle profile of BRCA2-deficient cells during the transition from acute to chronic responses (Fig. 2e). Initially cells accumulated in G1, with levels of S and G2/M cells correspondingly decreased, indicative of G1 cell cycle arrest in the short term after BRCA2 inactivation. This correlates with transcriptional downregulation of genes controlling cell cycle and DNA replication, as demonstrated by RNA-seq data (Fig. 2a, c). The frequency of G1 cells reached a peak at day 12 and subsequently diminished, without however reaching the level of BRCA2-proficient cells. The percentage of S-phase cells remained lower in cells lacking BRCA2 relative to wild type for the course of the experiment. These results are consistent with the observation that the proliferation rate of BRCA2-deficient cells is lower than that of BRCA2-proficient control cells (Supplementary Fig. 4).

Upon long-term BRCA2 inactivation, RNA-seq analyses revealed upregulation of innate immune response genes (Fig. 2b, d). A connection between innate immune response induction and genome instability was established by recent studies[19–21]. Loss of chromosome integrity leads to cytosolic DNA accumulation in the form of micronuclei, which bind and activate the cytosolic DNA sensor cGAS, a key player in interferon signaling activation. We thus evaluated the levels of cGAS-positive micronuclei in BRCA2-deficient cells and their dynamics over the course of the experiment (Fig. 2f). We observed the highest frequency of cGAS-positive micronuclei upon chronic BRCA2 inactivation (28 days of DOX addition), which correlated with the upregulation of innate immune response genes. This suggests that micronuclei formation and cGAS activation could mediate the immune responses triggered by BRCA2 abrogation.

Differential gene expression analyses were also conducted in MDA-MB-231 cells carrying a DOX-inducible BRCA2 shRNA (Supplementary Fig. 6a). These revealed only 19 genes downregulated (Log$_2$(Fold Change) < −0.5; Supplementary Data 1) at day 4 and 91 genes upregulated (Log$_2$(Fold Change) > 0.5; Supplementary Data 2) at day 28 of DOX treatment.

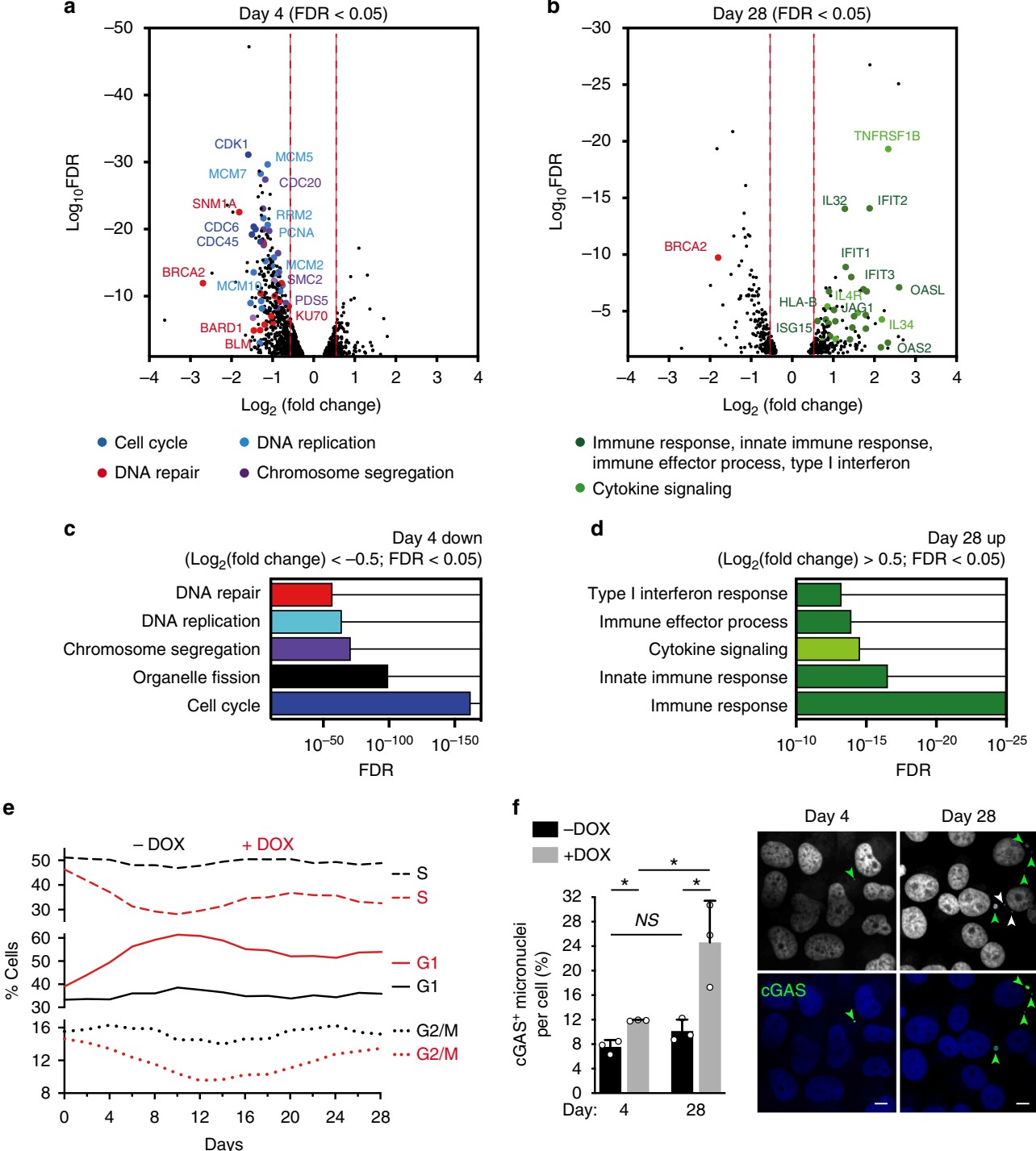

**Fig. 2** Pathway deregulation in BRCA2-deficient H1299 cells during short-term (4 days) or long-term (28 days) *BRCA2* inactivation. **a**, **b** Volcano plot of genes differentially expressed (FDR < 0.05) in BRCA2-deficient versus BRCA2-proficient H1299 cells, after DOX treatment for 4 (**a**) or 28 (**b**) days. A subset of the genes significantly downregulated (Log$_2$(Fold Change) < −0.5) after 4 days or significantly upregulated (Log$_2$(Fold Change) > 0.5) after 28 days of DOX treatment is shown. **c**, **d** Gene set enrichment analysis based on functional annotation (Gene Ontology—Biological Process database) of genes downregulated after 4 days (**c**) or upregulated after 28 days (**d**) of DOX treatment. **e** Cells were pulse-labeled with EdU for 30 min. Frequency of cells in G1, S, and G2/M stages of the cell cycle were determined using FACS analyses of EdU-labeled cells. **f** Cells were fixed and stained with antibody against cGAS after 4 days or after 28 days of DOX treatment. DNA was counterstained with DAPI. Shown are representative images of cells treated with DOX. cGAS-positive micronuclei were quantified and related to number of cells. Error bars represent SD of $n = 3$ independent experiments. *NS*, $p > 0.05$; *, $p < 0.05$ (unpaired two-tailed *t* test). Green arrows indicate cGAS-positive micronuclei and white arrows indicate cGAS-negative micronuclei. Scale bar represents 10 µM

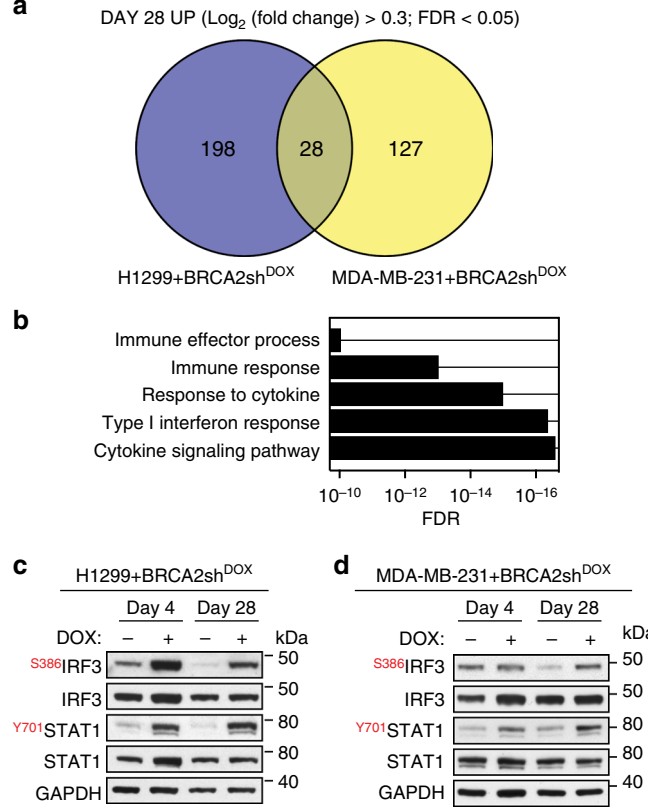

**Fig. 3** Upregulation of innate immune response genes in BRCA2-deficient human cells. **a** Venn diagram of common genes upregulated in H1299 and MDA-MB-231 human cells, upon chronic (28 days) DOX-induced BRCA2 depletion. **b** Gene set enrichment analysis (Gene Ontology—Biological Process database) of the 28 genes upregulated upon chronic BRCA2 inactivation in both H1299 and MDA-MB-231 cells. **c**, **d** Whole-cell extracts prepared after 4 and 28 days of DOX treatment were immunoblotted as indicated. GAPDH was used as a loading control. Phosphorylation site are indicated in red

REACTOME biological pathway analyses showed enrichment in processes, including α, β-interferon signaling, and cytokine signaling in the immune system (Supplementary Fig. 6b), and were therefore analogous to the processes upregulated in H1299 cells after 28 days of DOX treatment.

Moreover, we performed RNA-seq analyses of differential gene expression in $BRCA2^{-/-}$ versus $BRCA2^{+/+}$ DLD1 cells (Supplementary Fig. 7). Pathway deregulation score analyses[22] of the REACTOME database using the top upregulated genes in $BRCA2^{-/-}$ DLD1 cells identified interferon signaling, cytokine signaling, and immune response as the highest scoring pathways.

**Induction of immune response genes in BRCA2-deficient cells.** Long-term BRCA2 inactivation in H1299 and MDA-MB-231 human cells led to upregulation of common immune processes. To facilitate experimental validation for these findings, we identified common genes induced in both cell lines after 28 days of DOX treatment (Fig. 3a). We intersected the list of genes significantly upregulated in H1299 and MDA-MB-231 cells at day 28 (FDR < 0.05) and further filtered the high-confidence hits using Log2(Fold Change) > 0.3. This analysis identified 28 genes upregulated in both cell lines (Table 1), which showed enrichment in gene ontology processes, including cytokine signaling, type I interferon response and immune response (Fig. 3b).

In BRCA2-deficient cells, replication fork instability and nucleolytic degradation inflict replication-associated DNA damage, which conceivably cause micronuclei accumulation and activate cGAS-dependent innate immune responses (Fig. 2f). Phosphorylation of STAT1 at Tyr701 is routinely used as a marker for activation of cytokine signaling[21], including interferon type I[23] in response to cytosolic DNA. Consistent with this, we observed enhanced STAT1 Tyr701 phosphorylation (Fig. 3c) in BRCA2-deficient relative to BRCA2-proficient H1299 cells, after 4 or 28 days of DOX treatment (Fig. 3c). IRF3 phosphorylation at Ser386, indicative of its nuclear translocation[24], was also induced by BRCA2 depletion. We detected a similar response upon BRCA2 inactivation in MDA-MB-231 cells (Fig. 3d).

Thus, BRCA2 inhibition in human cells activates DNA damage responses and stimulates innate immune gene expression, as previously reported for DNA damaging agents[21,23] and SAMHD1 inactivation[20].

**Cytosolic DNA accumulation triggers innate immune responses.** We next used quantitative RT-PCR to validate the induction of interferon response genes identified with RNA-seq analyses. We evaluated the dynamics of immune response gene expression by monitoring changes in mRNA levels over a 28-day period (Fig. 4a). mRNA levels of ISGs (*IFIT1*, *IFIT2*, *IFIT3*, *IFI6*, *OAS1*, *OAS2*, and *ISG15*) and cytokine signaling gene (*TNFRSF1B*) were measured at 2-day intervals, both in BRCA2-proficient and -deficient cells. Expression of all genes monitored in this manner increased gradually during chronic BRCA2 inactivation in H1299 cells. Cytosolic DNA accumulated in the same temporal manner, suggesting a correlation between the two events.

Additionally, DOX-induced BRCA2 depletion triggered STAT1 phosphorylation at Tyr701 and IRF3 phosphorylation at Ser386 (Fig. 4b). These phosphorylation events preceded upregulation of the innate immune response genes, consistent with the roles of IRF3 and STAT1 in promoting transcription of the ISGs. Interestingly, STING protein levels were also markedly increased upon BRCA2 abrogation, while no significant change was detected in its mRNA levels. This may reflect stabilization of this protein, possibly by inhibition of its proteolytic degradation[25].

Short-hairpin RNA (shRNA)-mediated interference is known to induce innate immune responses[26]. In order to assess whether the immune response detected in our system is due to BRCA2 abrogation per se or to non-specific shRNA accumulation, we evaluated interferon gene expression in two additional cell lines carrying DOX-inducible shRNAs against ERK1 or ERK2 (Supplementary Fig. 8). Expression of shRNAs targeting BRCA2, ERK1, or ERK2 was induced with DOX over a 28-day period. The mRNA levels of ISGs *IFIT1*, *IFIT2*, *OAS1*, or *OAS2* were measured using quantitative RT-PCR at day 4 and day 28, in DOX-treated and untreated control cells (Supplementary Fig. 8a). We observed upregulated immune gene expression only upon BRCA2 shRNA induction. ERK1 and ERK2 shRNA were effectively induced, as shown by reduction in protein levels detected in immunoblot analyses (Supplementary Fig. 8b). Moreover, IRF3 or STAT1 phosphorylation, a signature of interferon signaling activation, was only detected in cells expressing BRCA2 shRNA (Supplementary Fig. 8b). These results indicate that the innate immune response is specifically triggered by loss of BRCA2 and not as a consequence of shRNA accumulation.

We further investigated whether the cGAS-STING pathway was critical for induction of innate immune responses upon loss of BRCA2. Inhibition of STING using siRNA suppressed upregulation of the ISGs (Supplementary Fig. 9a) and

**Table 1 List of 28 genes commonly upregulated in H1299 and MDA-MB-231**

| Gene | H1299 | | | | MDA-MB-231 | | | |
|---|---|---|---|---|---|---|---|---|
| | LFC_d4 | FDR_d4 | LFC_d28 | FDR_d28 | LFC_d4 | FDR_d4 | LFC_d28 | FDR_d28 |
| IFIT1* | −0.37 | 7.3E−02 | 1.44 | 9.7E−09 | −0.32 | 4.2E−01 | 0.71 | 2.5E−05 |
| IFIT2* | −0.04 | 9.3E−01 | 1.89 | 8.3E−15 | −0.39 | 2.7E−01 | 0.70 | 5.1E−04 |
| IFIT3* | −0.13 | 6.7E−01 | 1.30 | 1.2E−09 | −0.23 | 3.7E−01 | 0.47 | 1.1E−03 |
| OAS2* | 0.43 | NA | 2.16 | 1.5E−02 | 0.23 | 8.5E−01 | 0.58 | 4.2E−02 |
| OASL* | 0.06 | 9.4E−01 | 2.60 | 7.7E−08 | −0.04 | NA | 1.19 | 1.7E−02 |
| ISG15* | −0.19 | 4.5E−01 | 0.88 | 1.2E−04 | −0.07 | 9.5E−01 | 0.40 | 8.1E−04 |
| IL13RA1* | 0.33 | 4.3E−02 | 0.48 | 2.4E−02 | 0.25 | 2.0E−01 | 0.40 | 2.6E−03 |
| IL27RA* | 0.51 | 8.7E−03 | 0.60 | 1.7E−02 | 0.15 | 8.4E−01 | 0.53 | 2.7E−04 |
| IL4R* | 0.31 | 1.4E−01 | 0.86 | 3.8E−06 | 0.10 | 8.7E−01 | 0.37 | 2.1E−03 |
| JAG1* | 0.38 | 1.1E−01 | 1.02 | 7.9E−06 | 0.22 | 1.0E−01 | 0.46 | 1.3E−08 |
| HLA-B* | 0.07 | 7.8E−01 | 0.62 | 7.2E−05 | 0.10 | 7.2E−01 | 0.40 | 2.0E−05 |
| HLA-C* | 0.09 | 7.0E−01 | 0.51 | 7.0E−03 | 0.06 | 9.4E−01 | 0.30 | 2.0E−02 |
| HMOX1* | 0.49 | 1.4E−03 | 0.48 | 4.8E−02 | −0.09 | 9.5E−01 | 0.92 | 4.2E−02 |
| QPCT | 0.06 | 8.6E−01 | 0.66 | 2.1E−02 | 0.41 | 4.8E−01 | 0.67 | 6.6E−03 |
| SLC46A3 | 0.20 | 5.1E−01 | 1.17 | 1.5E−07 | 0.05 | 9.9E−01 | 0.56 | 2.6E−02 |
| SMPD1 | 0.51 | 1.6E−03 | 0.84 | 7.9E−05 | 0.22 | 5.3E−01 | 0.43 | 4.5E−03 |
| NEU1 | 0.23 | 1.7E−01 | 0.59 | 3.4E−03 | 0.08 | 9.1E−01 | 0.35 | 9.8E−04 |
| ABCA1 | 0.78 | 8.0E−04 | 0.85 | 1.6E−02 | 0.54 | NA | 0.94 | 1.9E−02 |
| ACSL5 | 0.50 | NA | 1.54 | 6.1E−04 | 0.19 | 1.8E−01 | 0.33 | 4.3E−03 |
| ADGRG1 | 0.83 | 2.1E−06 | 0.79 | 5.3E−04 | 0.48 | 2.1E−02 | 0.70 | 2.2E−06 |
| CALB2 | 0.24 | 4.5E−01 | 0.73 | 1.6E−02 | 0.28 | 2.0E−01 | 0.43 | 4.5E−02 |
| CDKN1A | 0.05 | 8.9E−01 | 0.55 | 6.7E−03 | 0.13 | 7.8E−01 | 0.64 | 4.5E−02 |
| CHPF | 0.61 | 7.2E−06 | 1.05 | 1.0E−08 | 0.26 | 6.9E−02 | 0.57 | 5.4E−08 |
| COL6A1 | 0.37 | 1.9E−03 | 0.40 | 3.7E−02 | 0.25 | 4.5E−01 | 0.71 | 9.7E−09 |
| ENG | 0.69 | 5.4E−03 | 1.05 | 2.9E−03 | 0.20 | 7.4E−01 | 0.63 | 8.6E−05 |
| FBLIM1 | 0.58 | 1.0E−03 | 1.22 | 4.1E−06 | −0.01 | NA | 1.30 | 2.3E−02 |
| HIST1H2BD | 0.23 | 3.4E−01 | 0.73 | 4.5E−03 | −0.14 | 9.2E−01 | 0.52 | 1.3E−02 |
| ID3 | −0.23 | 2.1E−01 | 0.58 | 2.2E−02 | 0.17 | 8.4E−01 | 0.37 | 8.7E-03 |

Asterisk indicates cytokine signaling, type I interferon response and immune response genes; LFC, Log (Fold Change)

correspondingly attenuated phosphorylation of IRF3 or STAT1 in BRCA2-depleted cells (Supplementary Fig. 9b). Consistent with the concept that cGAS-STING activation and interferon signaling triggers JAK/STAT-dependent gene expression, we found that siRNA-mediated STAT1 depletion decreased the mRNA levels of ISGs (Supplementary Fig. 9c). The effect of STAT1 abrogation was relatively mild due to the fact that cells were treated with STAT1 siRNA after 28 days of DOX treatment. In contrast, treatment with STING siRNA commenced concomitantly with DOX addition.

**PARP inhibitors potentiate the innate immune response in vitro and in vivo.** To establish the clinical relevance of our results, we investigated the immune response gene expression in BRCA2-deficient ovarian serous cystadenocarcinomas using mRNA expression data available in The Cancer Genome Atlas (TCGA; Fig. 5). Differential expression analysis was performed on the subset of BRCA2-deleted ($BRCA2^{del}$, $n = 4$; Fig. 5a) and RAD51-deleted ($RAD51^{del}$, $n = 11$; Fig. 5b) tumors relative to cases with median BRCA2 or RAD51 mRNA levels ($BRCA2^{median}$ or $RAD51^{median}$, $n = 145$). Out of the list of eight genes validated by quantitative RT-PCR, the seven ISGs (IFIT1, IFIT2, IFIT3, IFI6, OAS1, OAS2, and ISG15) had higher mRNA levels in $BRCA2^{del}$ or $RAD51^{del}$ compared with control tumors. Similar upregulation of these genes was found in tumors carrying deletions of HR genes BRCA1, BRCA2, RAD51, and PALB2 relative to tumors with median HR gene mRNA levels within the TCGA cohort (Supplementary Fig. 10). These results suggested that upregulation of this subset of innate immune response genes upon chronic inactivation of BRCA2 observed in vitro is recapitulated in HR-deficient ovarian human tumors.

Notably, four of the ISGs (IFIT1, IFIT2, OAS2, and ISG15) were also upregulated in the MDA-MB-231 cells lacking BRCA2 after 28 days of DOX treatment (Table 1), albeit at lower levels than in H1299 cells. This result may reflect transcriptional regulation specific to the breast tumor from which this cell line originates.

Small molecule inhibitors of poly[ADP-ribose] polymerase (PARP) have been recently approved for clinical treatment of BRCA1- and BRCA2-deficient ovarian and breast cancers[27,28]. The molecular mechanisms of selective targeting of BRCA-deficient cells and tumors by PARP inhibitors has been investigated extensively. Trapping of the PARP1 on DNA ends obstructs replication fork progression leading to accumulation of single strand breaks, which are converted into lethal DSB lesions[29–32].

Induction of DNA damage by PARP inhibitors prompted us to investigate whether treatment of BRCA2-deficient cells with these drugs could also impact on the interferon response. To address this, we treated H1299 cells with DOX for 4 days to induce BRCA2 depletion. Under these conditions, we did not detect upregulation of ISGs using quantitative RT-PCR (Fig. 4a). However, incubation with 1 or 10 μM olaparib for 72 h induced a dose-dependent increase in the mRNA levels of ISGs (Fig. 5c). Consistent with these results, olaparib also triggered phosphorylation of IRF3 and STAT1 in a dose-dependent manner in the BRCA2-deficient cells (Fig. 5d). Surprisingly, the same phosphorylation events were detected in BRCA2-proficient cells treated with 10 μM olaparib. We thus monitored DNA damage induction through detection of KAP1 phosphorylation at S824, a well-characterized marker for DSB and ATM activation (Fig. 5d). Treatment with 10 μM, but not 1 μM olaparib triggered KAP1 phosphorylation in BRCA2-proficient cells, while DNA damage

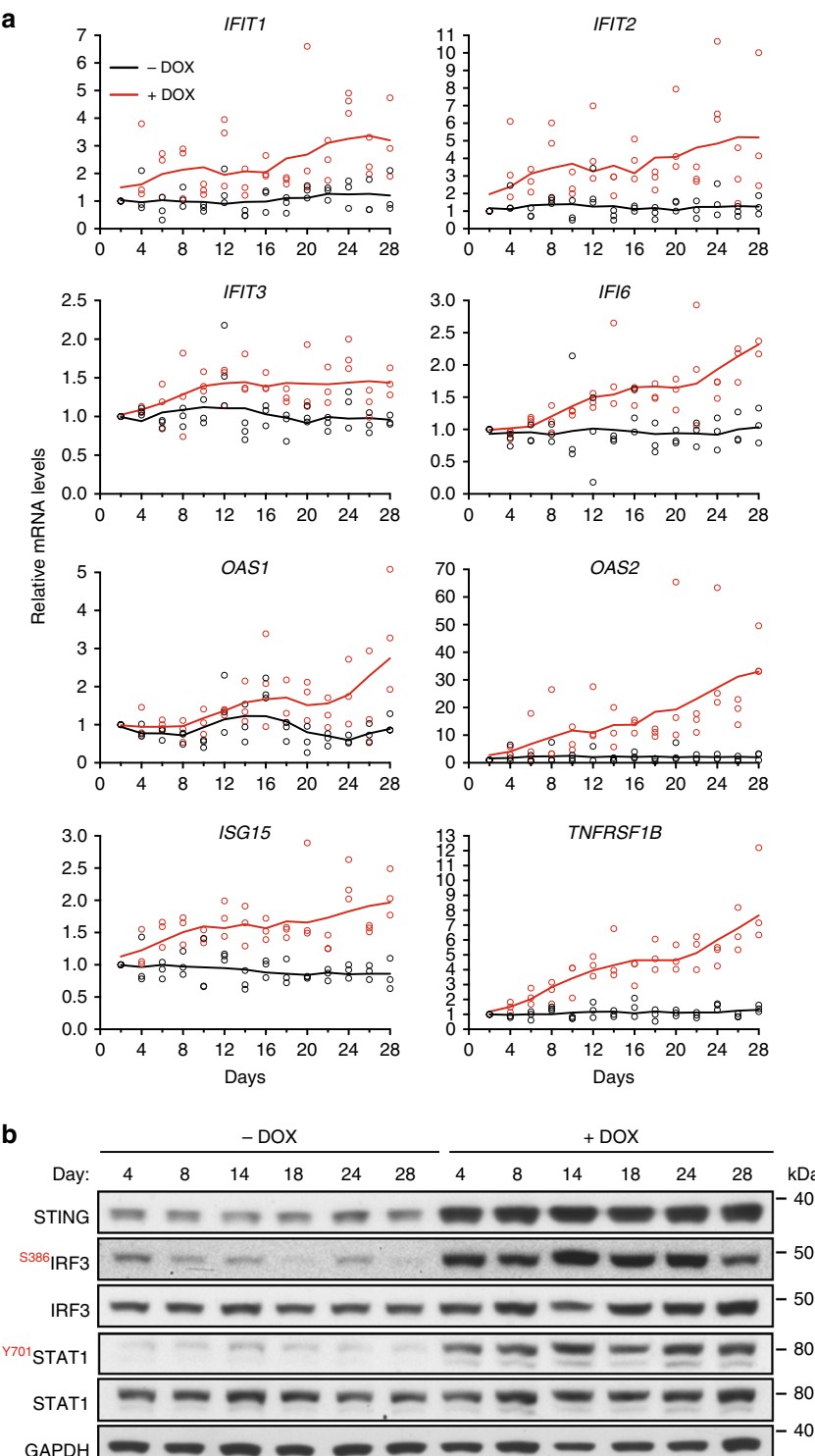

**Fig. 4** Time course of innate immune response activation in BRCA2-deficient cells. **a** H1299 cells expressing a DOX-inducible BRCA2 shRNA were grown in the presence or absence of DOX for 28 days. Cells collected every 2 days were subjected to quantitative RT-PCR analyses using primers specific for indicated genes. mRNA levels were expressed relative to the gene encoding β-actin and to day 2 ($2^{-\Delta\Delta CT}$). $n = 3$ independent experiments, each performed in triplicates. Each dot represents a single replicate. **b** Whole-cell extracts prepared at indicated times after DOX addition were immunoblotted as shown. GAPDH was used as a loading control. Phosphorylation sites are indicated in red

induction was detected in BRCA2-deficient cells treated with both 1 and 10 μM olaparib (Fig. 4d). We concluded that BRCA2-deficient cells are more susceptible to DNA damage induced by olaparib, which mediates rapid activation of the innate immune responses, even after 4 days of BRCA2 depletion.

We next monitored the impact of olaparib treatment on the cell cycle profile of BRCA2-proficient and -deficient cells (Fig. 5e). As previously reported[4], BRCA2 inactivation for 4 days promoted cell cycle arrest in G1 and reduced the frequency of cells in S and G2/M stages of the cell cycle. We observed that olaparib

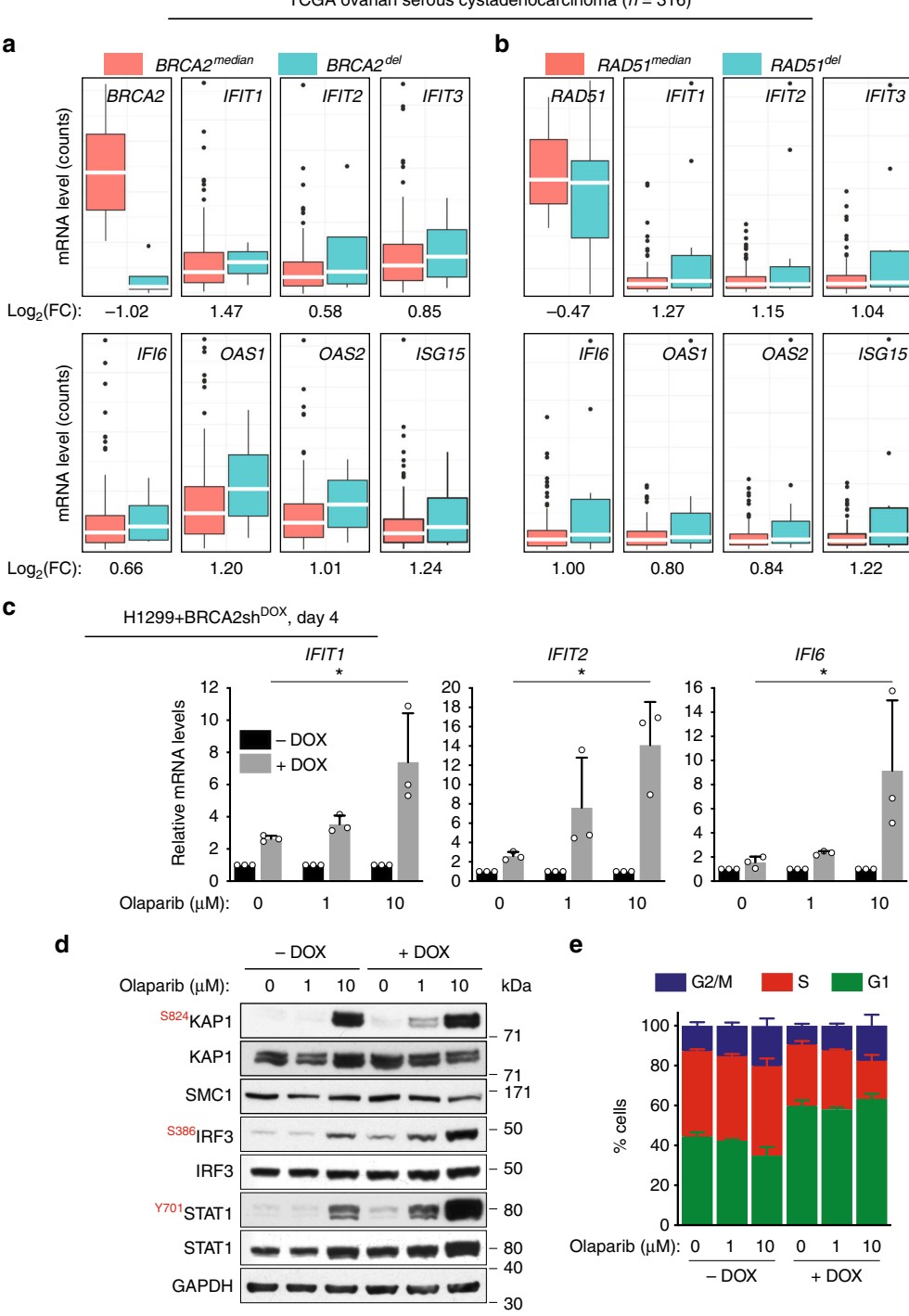

**Fig. 5** ISG induction in HR-deficient human cells and tumors and olaparib impact on this process. **a** Upregulation of innate immune response genes in *BRCA2*-deleted ovarian tumors (*n* = 4) versus tumors with median *BRCA2* mRNA expression (*n* = 145). **b** Upregulation of innate immune response genes in *RAD51*-deleted ovarian tumors (*n* = 11) versus tumors with median *RAD51* mRNA expression (*n* = 145). Dots in graphs represent individual tumors. Middle line (white), median; box limits 25 and 75 percentiles; whiskers, minimum and maximum values. **c** H1299 cells expressing a DOX-inducible BRCA2 shRNA were grown in the presence or absence of DOX for 4 days. Then, olaparib (1 or 10 μM) was added for 72 h, followed by quantitative RT-PCR analyses. Primers specific for the indicated genes were used. mRNA levels were expressed relative to the gene encoding β-actin and to untreated (−DOX) control cells ($2^{-\Delta\Delta CT}$). Error bars represent SD of *n* = 3 independent experiments. *, *p* < 0.05 (unpaired two-tailed *t* test). **d** Whole-cell extracts from cells treated as in (**c**) were immunoblotted as shown. GAPDH and SMC1 were used as loading controls. Phosphorylation sites are indicated in red. **e** Cells treated as in (**c**) were pulse-labeled with EdU for 30 min. Frequency of cells in G1, S, and G2/M stages of the cell cycle were determined using FACS analyses of EdU incorporation and propidium iodide staining. Error bars represent SD of *n* = 3 independent experiments

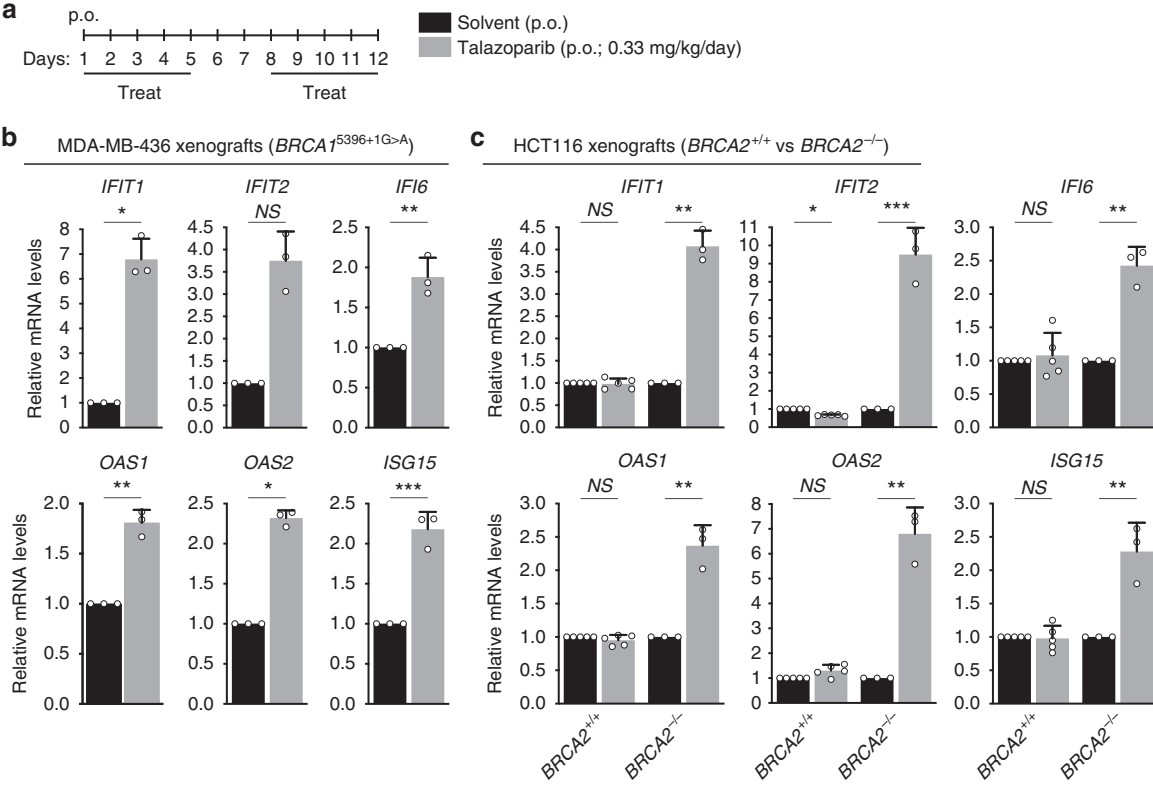

**Fig. 6** Effect of PARP inhibitor on ISG expression in BRCA1/2-deficient tumors. **a** Schematic representation of tumor treatment. Tumor-bearing mice were treated on indicated days with PARP inhibitor talazoparib or with solvent control, both orally administered (p.o.). Total RNA was prepared from tumors collected at the end of the treatment. **b** BRCA1-deficient MDA-MB-436 human cells ($1 \times 10^6$) were injected into the mammary fat pad and **c** HCT116 ($BRCA2^{+/+}$ or $BRCA2^{-/-}$) human cells ($3 \times 10^6$) were injected intramuscularly into the hind leg muscles of CB17/SCID mice and allowed to generate tumors which were then treated as in (**a**). mRNA levels of indicated genes were determined using quantitative RT-PCR analyses and were expressed relative to the gene encoding β-actin (**b**) or GAPDH (**c**) and to solvent-treated control tumors ($2^{-\Delta\Delta CT}$). Error bars represent SD of $n = 3$ (**b**, **c** $BRCA2^{-/-}$) or $n = 5$ (**c**, $BRCA2^{+/+}$) tumors, for which quantitative RT-PCR reaction was performed in triplicates. NS, $p > 0.05$; *, $p < 0.05$; **, $p < 0.01$; ***, $p < 0.001$ (unpaired two-tailed $t$ test)

treatment did not affect G1, however, it increased the proportion of BRCA2-deficient cells in G2/M. In particular, treatment with 10 μM olaparib reduced the S-phase cell population, while increasing the frequency of cells in G2/M. This is consistent with a previous study[33], which reported that treatment of BRCA1-deficient cells with 10 μM olaparib accelerates replication fork progression leading to G2/M accumulation.

To address whether the innate immune response triggered by BRCA inactivation in vitro can be recapitulated in vivo, in the tumor context, we established orthotopic xenograft tumors by injecting BRCA1-compromised MDA-MB-436 mammary breast cancer cells[34] into the mammary fat pad of CB17/SCID mice (Fig. 6a, b). These cells carry a 5396 + 1G > A mutation in BRCA1[34], which abrogates BRCA1 protein expression. Treatment with the PARP inhibitor talazoparib triggered ISG expression in these BRCA1-deficient tumours, as demonstrated by quantitative RT-PCR analysis of tumor RNA.

We established a second xenograft model using isogenic $BRCA2^{+/+}$ and $BRCA2^{-/-}$ human colorectal carcinoma HCT116 cells[35] (Fig. 6c). ISGs were upregulated specifically in tumors derived from $BRCA2^{-/-}$, but not $BRCA2^{+/+}$ cells, upon talazoparib treatment. These results support the notion that PARP inhibitors elicit an innate immune response in BRCA1- or BRCA2-deficient xenograft tumors.

## Discussion
Although not lethal, BRCA2 inactivation in cancer-derived human cells significantly slows down cell proliferation. This

indicates that cells can adapt to loss of BRCA2 and to the perturbations in cell physiology this entails. Here we explored the changes in transcriptome associated with BRCA2 abrogation and defined distinct transcriptional alterations, evolving from an early, acute response, to a late, chronic response. The former was characterized primarily by decreased mRNA levels of genes required for cell cycle progression, chromosome segregation, DNA repair and replication, consistent with previously reported roles for BRCA2 in replication fork stability and DNA repair[1]. EdU incorporation analyses revealed that these transcriptional changes are associated with accumulation of BRCA2-deficient cells in G1 and with low levels of entry/progression through S and G2/M phases of the cell cycle. Slowing down these processes may be conducive for activation of alternative, BRCA2-independent mechanisms of fork protection and re-start, which, in the long-term, sustain cell survival.

The late, chronic response to BRCA2 inactivation consisted of upregulation of interferon-regulated innate immune response genes. Initially identified as a mechanism of defense against pathogens by blocking viral infections and priming adaptive immune responses[36], the innate immune response can also be enlisted to counteract tumor progression[14]. Genomic DNA is evicted into the cytoplasm as a consequence of the genomic instability intrinsic to most tumors. Cytoplasmic DNA assembles into micronuclei where it is recognized as pathogenic by the DNA sensor cGAS, which triggers STING activation (Fig. 7). This, in turn, facilitates phosphorylation and nuclear translocation of interferon response factors (e.g., IRF3), which drive transcription of the interferon response genes. Secreted

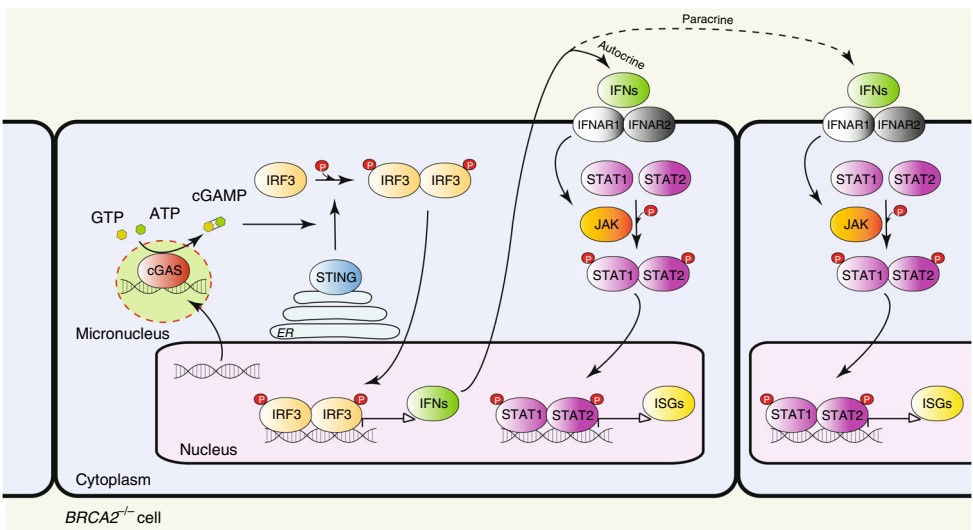

**Fig. 7** Model of innate immune response activation in BRCA2-deficient cells. Intrinsic replication stress and unrepaired DSBs cause genomic instability leading to release of DNA into the cytosol. Cytosolic DNA assembles into micronuclei, where it binds to and activates cyclic GMP-AMP synthase (cGAS), a cytosolic DNA sensor. cGAS catalyzes the synthesis of cyclic GMP-AMP (cGAMP), which binds to and activates STING (stimulator of interferon genes). STING promotes TBK1-dependent phosphorylation of interferon response factor 3 (IRF3), which translocates into the nucleus as a homodimer to induce transcription of type I interferon. Secreted interferon binds to transmembrane receptors and elicits autocrine and paracrine signaling, both mediated by the JAK-STAT pathway. Janus kinases (JAKs) phosphorylate signal transducer and activator of transcription 1 (STAT1) and 2 (STAT2), which enter the nucleus as a heterodimer to initiate transcription of interferon-stimulated genes (ISGs). ER, endoplasmic reticulum

interferon acts similarly to other cytokines to activate signaling pathways, in particular the JAK/STAT1 pathway, within the same cell (autocrine response) or in adjacent cells (paracrine response), resulting in ISG induction. ISG upregulation can also be triggered by ionizing radiation[21] and chemotherapeutic drugs (e.g., cisplatin, MMC)[23] and requires passage through mitosis with associated micronuclei formation[19,21].

Here we demonstrate that the mRNA levels of ISGs are enhanced by chronic BRCA2 inactivation. We previously reported that severe replication stress[5] and persistent DNA damage associated with loss of BRCA2 result in aberrant chromosome segregation[4]. The data presented here show that chronic BRCA2 abrogation leads to micronuclei formation, which recruit the cytosolic DNA sensor cGAS. Conceivably, cGAS activation triggered interferon signaling, as suggested by the phosphorylation of STAT1 and IRF3, which led to induction of ISG expression. Consistent with this hypothesis, transcriptional upregulation of these genes was STING- and STAT1-dependent (Supplementary Fig. 9) and STAT1 Tyr701 phosphorylation, required for its nuclear translocation and activation, is detectable from the early stages of BRCA2 depletion (Fig. 5b).

How the innate immune response activation impacts on the long-term survival of BRCA2-deficient cells and tumors remains to be elucidated. In our in vitro system, the interferon response induced upon chronic BRCA2 inactivation is associated with partial restoration of cell cycle progression (Fig. 5c). Upregulation of immune signaling factors known to promote cell proliferation (e.g., TNFRSF1B, also known as TNFR2 or p75[37]) could account for the ability of BRCA2-deficient cells to survive. Prolonged cell survival in the absence of BRCA2 may potentiate the innate immune response, as successive rounds of DNA replication and chromosome segregation are likely to augment their genomic instability. Thus, that the early and late responses to BRCA2 inactivation are linked to each other: replication stress could slow down cell cycle progression and suppress the immune gene upregulation during the early stages of BRCA2 abrogation, while replication stress adaptation promotes cell cycle re-entry and triggers the immune response during the late stages.

In the tumor context, ensuing interferon responses may stimulate cytotoxic T-cell activation, analogous to the STING-dependent defense mechanisms elicited by viral infection[38], which could be onco-suppressive. Studies using mouse models of tumorigenesis driven by *Brca2* abrogation[39] must be conducted in order to evaluate this possibility. Conversely, it will be interesting to determine whether BRCA2-deficient tumors that became resistant to chemotherapy have also acquired the ability to escape immune detection.

A recent study has shown that PARP inhibitors inflict DNA damage and promote cGAS/STING pathway activation in vitro, independently of BRCA status[40]. Our results establish that olaparib accelerates upregulation of innate immune response genes specifically in BRCA1- or BRCA2-deficient tumors. Conceivably, the intrinsically high levels of DNA damage in BRCA2-deficient cells were further increased by olaparib, which thereby potentiated the innate immune response. In our study, the BRCA2-proficient xenograft tumors failed to mount an innate immune response upon PARP inhibitor treatment. This inconsistency with a previous study[40] possibly reflects differences between the tumor models and/or the length of PARP inhibitor treatment.

Our discovery that the chronic response to BRCA2 inactivation is associated with innate immune response upregulation, potentiated by PARP inhibitors, has important therapeutic implications. First, it predicts that drugs that specifically kill BRCA2-deficient cells and tumors by inflicting DNA damage, may concomitantly trigger cytotoxic immune responses. To what extent the deleterious effects of these drugs entail immune response activation remains to be established. Second, it is possible that BRCA-compromised tumors, which become olaparib-resistant through restoration of DNA repair[41–43] lose the capacity to mount innate immune responses. This tumor subset may be effectively targeted through therapeutic approaches that restore immune signaling.

## Methods
**Cell lines and growth conditions**. Human non-small cell lung carcinoma H1299 cells and human invasive ductal breast cancer MDA-MB-231 cells, wild type (American Type Culture Collection) or carrying a doxycycline (DOX)-inducible

BRCA2 shRNA[4], were cultivated in monolayers in DMEM medium (Sigma) supplemented with 10% tetracycline free fetal bovine serum (Clontech). For induction of shBRCA2, 2 µg/mL DOX (D9891, Sigma) was added to growth medium. Human colorectal adenocarcinoma DLD1 cells, parental and BRCA2-mutated (Horizon Discovery[3]), were cultivated in monolayers in DMEM medium (Sigma) supplemented with 10% fetal bovine serum (Thermo Fisher Scientific) and 1% penicillin/streptomycin (Sigma).

**Immunoblotting**. To prepare whole-cell extracts, cells were washed once in 1xPBS, harvested by trypsinization, washed in 1xPBS and re-suspended in SDS-PAGE loading buffer, supplemented with 0.1 mM DTT. Samples were sonicated and heated at 70 °C for 10 min. Equal amounts of protein (50–100 µg) were analyzed by gel electrophoresis followed by western blotting. NuPAGE-Novex 4–12% Bis-Tris and NuPAGE-Novex 3–8% Tris-Acetate gels (Life Technologies) were run according to manufacturer's instructions. Uncropped scans are provided in the Source Data file.

**RNA-seq analyses**. H1299, H1299+ shBRCA2$^{DOX}$, MDA-MB-231, and MDA-MB-231+ shBRCA2$^{DOX}$ were cultured in presence or absence of DOX for 4 or 28 days. Cells were collected for RNA extraction and processed using the RNeasy® Mini Kit (Qiagen, #74104) according to the manufacturer's guidelines. RNA samples were quantified using RiboGreen (Invitrogen) on the FLUOstar OPTIMA plate reader (BMG Labtech) and the size profile and integrity analyzed on the 2200 or 4200 TapeStation (Agilent, RNA ScreenTape). RNA integrity number estimates for all samples were between 9.0 and 10.0. Input material was normalized to 1 µg prior to library preparation. Poly(A) transcript enrichment and strand specific library preparation were performed using TruSeq Stranded mRNA kit (Illumina) following manufacturer's instructions. Libraries were amplified (15 cycles) on a Tetrad (Bio-Rad) using in-house unique dual indexing primers[44]. Individual libraries were normalized using Qubit and size profile was analyzed on the 2200 or 4200 TapeStation. Individual libraries were pooled together and pooled libraries were diluted to ~10 nM for storage. Each library aliquot was denatured and further diluted prior to loading on the sequencer. Paired end sequencing was performed using a HiSeq4000 75 bp platform (Illumina, HiSeq 3000/4000 PE Cluster Kit and 150 cycle SBS Kit), generating a raw read count of > 22 million reads per sample.

**RNA-seq data processing**. Reads were aligned to the human reference genome (GRCh37) using HISAT2[45] and duplicate reads removed using the Picard Mark-Duplicates tool (Broad Institute). Reads mapping uniquely to Ensembl-annotated genes were summarised using featureCounts[46]. The raw gene count matrix was imported into the R/BioConductor environment[47] for further processing and analysis. Genes with low read counts (less than ~10 reads in more than three samples) were filtered out, leaving sets of 14,000–15,000 genes to test for differential expression between conditions, depending on the samples considered.

**Differential expression analysis**. The analyses were carried out using the R package DESeq2 version 1.18.1[48]. Unless otherwise stated, differentially expressed genes were identified based on two criteria: FDR (false discovery rate using Benjamini-Hochberg adjusted $p$-values) < 0.05 and absolute value of Log$_2$(Fold Change) > 0.5. Differentially expressed genes were determined independently for the following human cell lines: DLD1 cells (parental vs BRCA2-mutated), H1299, MDA-MB-231, H1299 + shBRCA2$^{DOX}$ and MDA-MB-231 + shBRCA2$^{DOX}$, +DOX vs −DOX conditions, and two time points (4 days and 28 days). For each set of conditions, hierarchical clustering was performed for the top 500 differentially expressed genes using Euclidean distances. Intersections were used to identify differentially expressed genes common to various data sets.

**Gene set enrichment analysis**. The GO and REACTOME pathway enrichment analyses were performed using the Gene Set Enrichment Analysis (GSEA) software[49] with the Molecular Signatures Database collection. The NetworkAnalyst platform[17] and the Cytoscape 3.5 software[50] were used to visualize the first-order protein–protein interactions networks, based on the high-confidence (confidence score > 0.9 and experimental evidence required) STRING interactome database[18].

**Pathway deregulation scores**. The Pathifier algorithm[22] calculates a pathway deregulation score (PDS) based on gene expression matrices for each sample. PDS represents the extent to which the activity of a given pathway differs in a particular test sample from the activity in the matched control. Gene sets were retrieved from the REACTOME database[51] in GMT format and the R Bioconductor pathifier package version 1.16.0, along with custom R scripts were used to calculate deregulation scores of immune system related pathways in DLD1 BRCA2$^{−/−}$ cells.

**TCGA data analysis**. The Cancer Genome Atlas (TCGA) ovarian serous cysta-denocarcinoma annotated mutation files were retrieved from the cBioPortal database (http://www.cbioportal.org/)[52] for 316 samples. We analyzed mRNA expression by downloading 309 ovarian cases with RNA expression data using the R Bioconductor TCGAbiolinks package version 2.9.5[53]. Samples were split into BRCA2-deleted (BRCA2$^{del}$) and -median (BRCA2$^{median}$) groups. We determined

that there were four BRCA2 deep deletion cases within this cohort. The bulk samples were ranked according to BRCA2 mRNA levels, allowing us to define a BRCA2$^{median}$ subset of 145 samples. Differential expression analysis was performed for the BRCA2$^{del}$ ($n = 4$) versus BRCA2$^{median}$ ($n = 145$) tumor subset. Differential expression analysis was performed independently for RAD51$^{del}$ ($n = 11$) tumors versus RAD51$^{median}$ ($n = 145$) tumors.

In addition, we assessed differential expression in tumors carrying deletions of other genes encoding HR factors. Particularly, we found one BRCA1-deleted and one PALB2-deleted tumors and we analyzed these together with the four BRCA2-deleted and the eleven RAD51-deleted tumors within this cohort. For this analysis, samples were split into HR-deleted tumors ($n = 17$) and HR-proficient tumors ($n = 257$) with median BRCA1/BRCA2/PALB2/RAD51 mRNA expression levels.

**siRNAs**. Cells were transfected using Dharmafect 1 (Dharmacon). Briefly, $0.4$–$0.8 \times 10^6$ cells were transfected with 40 nM siRNA (ON-TARGET plus human STAT1 SMART pool, L-003543-00-0005 or TMEM173 SMART pool, L-024333-00-0005; Dharmacon) by reverse transfection in 6- or 10-cm plates.

**Cell viability assays for population doubling**. Cells were seeded in 96-well plates upon 3 or 27 days of DOX treatment. The next day (day 4 or day 28 after addition of DOX), viability was determined using resazurin-based assays. Cells were incubated with 10 µg/mL resazurin (R7017, Sigma-Aldrich) in growth medium at 37 °C for 2 hours. Fluorescence was measured (544 nm excitation and 590 nm emission) using POLARstar Omega plate reader (BMG Labtech). This was repeated at 2-day intervals and values were reported to the first measurement.

**Quantitative PCR (qPCR)**. For RNA reverse transcription, the Ambion Kit Power SYBR™ Green Cells-to-CT™ Kit (#4402954) was used according to manufacturer's instructions. Resulting complementary DNA was analyzed using SYBR Green technology in the QuantStudio 5 Real-Time PCR System. After normalization to $\beta$-actin or GAPDH, gene expression was calculated[54] relative to untreated control cells, as $2^{-\Delta\Delta CT}$. The following primer pairs were used: IFIT1, forward TAC CTG GAC AAG GTG GAG AA and reverse GTG AGG ACA TGT TGG CTA GA; IFIT2, forward TGT GCA ACC TAC TGG CCT AT and reverse TTG CCA GTC CAG AGG TGA AT; IFIT3, forward CTT CAG TAT TTA CTT GAG GCA GAC and reverse CTT GGT GAC CTC ACT CAT GAC; IFI6, forward TCG CTG ATG AGC TGG TCT GC and reverse ATT ACC TAT GAC GAC GCT GC; OAS1, forward CGC CTA GTC AAG CAC TGG TA and reverse CAG GAG CTC AGG GCA TA; OAS2, forward TCC AGG GAG TGG CCA TAG and reverse TCT GAT CCT GGA ATT GTT TTA AGT C; ISG15, forward GCG AAC TCA TCT TTG CCA GTA and reverse CCA GCA TCT TCA CCG TCA G; TNFRSF1B, forward CGG GAG CTC AGA TTC TTC CC and reverse GTC TCC AGC TGT GAC CGA AA; $\beta$-actin, forward ATT GGC AAT GAG CGG TTC and reverse GGA TGC CAC AGG ACT CCA T; GAPDH, forward GAC AGT CAG CCG CAT CTT CT and reverse ACC AAA TCC GTT GAC TCC GA.

**EdU incorporation in asynchronous cells**. To label replicated DNA, cells were incubated with 25 µM EdU for 30 min. Samples were collected by trypsinization and fixed using 90% methanol. Incorporated EdU was detected using the Click-iT EdU Alexa Fluor 647 Flow Cytometry Assay Kit (C10634, Invitrogen) according to manufacturer's instructions. Cells were re-suspended in 1xPBS containing 20 µg/mL propidium iodide (P4864, Sigma) and 400 µg/mL RNase A (12091-021, Invitrogen). Samples were processed using flow cytometry (BD FACSCalibur, BD Biosciences). 10000 events were analyzed per condition using FlowJo software.

**Antibodies**. The following antibodies were used for immunoblotting: mouse monoclonal antibodies raised against BRCA2 (1:1000, OP95, Calbiochem), GAPDH (1:30000, NB600-502, Novus Biologicals), α-Tubulin (1:30,000, TAT-1, Cancer Research UK Monoclonal Antibody Service); rabbit monoclonal antibody raised against ERK1/2 (1:5000, 4695, Cell Signaling), STING (1:1000, 13647, Cell Signaling), IRF3 (1:1000, AB76409, Abcam), phospho-IRF3 (1:1000, AB76493, Abcam), STAT1 (1:1000, 9175, Cell Signaling), phospho-STAT1 (1:1000, 9167, Cell Signaling); rabbit polyclonal antibodies raised against phospho-KAP1 (1:1000, A300-767A, Bethyl Laboratories), KAP1 (1:5000, A300-274A, Bethyl Laboratories), SMC1 (1:5000, A300-055A, Bethyl Laboratories).

**Immunofluorescence**. Cells on coverslips were washed in 1xPBS, swollen in hypotonic solution (85.5 mM NaCl and 5 mM MgCl$_2$) for 5 min and fixed with 4% paraformaldehyde for 10 min, before permeabilizing with 4% paraformaldehyde + 0.03% SDS. Cells were rinsed with 1xPBS + 0.4% Photo-Flo 200 (P7417-1PT, Sigma) and blocked with blocking buffer (1% goat serum, 0.3% BSA, 0.005% Triton X-100 in PBS). Then, cells were incubated with rabbit monoclonal antibody against cGAS (1:200, 15102, Cell Signaling) in blocking buffer overnight at room temperature. The next day, cells were washed and incubated with Alexa Fluor 488-conjugated secondary antibodies (1:400, A11008, Thermo) for 1 h at room temperature. Dried coverslips were mounted on microscope slides using ProLong Gold antifade reagent with DAPI (P36935, Thermo). Images were acquired using a Nikon Plan Apochromat 60 × 1.4 NA oil immersion lens on the Dragonfly imaging

system (Andor Technologies, Belfast UK) equipped with 405 and 488 nm lasers for visualizing DAPI and cGAS immunostained with an Alexa Fluor 488-conjugated antibody, respectively. The Dragonfly is built on a Nikon Ti2-E microscope body with the Nikon Perfect Focus System (Nikon Instruments, Japan). Data was collected using the Spinning Disk 40 μm pinhole mode of the Dragonfly on a Zyla 4.2 sCMOS camera using Fusion v2.0.0.15 software (Andor Technologies).

**In vivo xenograft experiments**. CB17/SCID mice (6 weeks old and weighing 26–28 g) were purchased from Charles River Laboratories (Calco, Italy). All animal procedures were in compliance with the national and international directives (D.L. March 4, 2014, no. 26; directive 2010/63/EU of the European Parliament and of the council; Guide for the Care and Use of Laboratory Animals, United States National Research Council, 2011).

MDA-MB-436 cells carrying $BRCA1^{5396+1G>A}$ mutation ($1 \times 10^6$) were injected into the mammary fat pad of CB17/SCID female mice. When the tumors were palpable (7 days after cell injection) the treatment was initiated.

To generate xenografts derived from HCT116 cells, wild type or carrying a CRISPR/Cas9 $BRCA2$ deletion[35], CB17/SCID male mice were injected intramuscularly into the hind leg muscles with $3 \times 10^6$ cells. When a tumor volume of ~250 mm³ was evident, the treatment was initiated. Each experimental group included five mice. Talazoparib (0.33 mg/kg/day) was administered orally for 5 consecutive days, followed by 2-day break and 5 more days of treatment. Tumors were excised and processed for RNA extraction using TRIzol reagent (Ambion, Life Technologies).

**Reporting summary**. Further information on research design is available in the Nature Research Reporting Summary linked to this article.

## Data availability
RNA-sequencing data are accessible at the Gene Expression Omnibus (GEO) repository, under accession number GSE123631. Other data from this study are available from the corresponding author upon request.

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

## Acknowledgements

We thank the Oxford Genomics Centre at the Wellcome Centre for Human Genetics (funded by Wellcome Trust grant reference 203141/Z/16/Z) for the generation and initial processing of the sequencing data. We thank John Broxholme and Eshita Sharma for developing and running the RNA-Seq data processing pipeline. We are grateful to Dr. Andrew Blackford (Department of Oncology, University of Oxford) for critical reading of the paper and to Dr. Johanna Michl (DPAG, University of Oxford) for help with RNA-seq analyses of DLD1 cells. Research in A.H. laboratory is supported by Breast Cancer Now and Breast Cancer Research Foundation. The work in A.B. laboratory was supported by Ministry of Health (Ricerca Corrente 2018) and Italian Association for Cancer Research (IG AIRC grant 21579). Work in A. L.-V. lab was supported by a PIC3i program from the Institut Curie (n° 91730 Prospects of Anticancer). E.P.L. is a recipient of a doctoral fellowship from the French Ministry of Education, Research and Technology. M.T.'s visit to A. L.-V. lab was supported by a Mayent-Rotschild fellowship from Institut Curie. Research in M.T. laboratory is supported by Cancer Research UK, Medical Research Council, University of Oxford, and EMBO Young Investigator Program. This project has received funding from the European Union's Horizon 2020 research and innovation program under the Marie Skłodowska-Curie grant agreement No. 722729.

## Author contributions

M.T., T.R. and E.P.L. designed the RNA-seq study. T.R. performed the RNA-seq experiments, B.W. and H.L. conducted the RNA-seq data alignment, E.P.L. and A.L.-V. conducted analyses of the RNA-seq data. T.R., A.M., and A.H. planned and performed validation of RNA-seq hits, including quantitative RT-PCR for time course analyses. F.G. performed time course analyses of EdU staining. F.G. and T.R. performed western blotting. M.P., S.D.V. and A.B. carried out the experiments involving the in vivo xenograft models. M.T. wrote the paper.

## Additional information

**Competing interests:** The authors declare no competing interests.

