## [Peer Review File · Nature Communications]

Reviewers' comments:

Reviewer #1 (Remarks to the Author):

This is an interesting manuscript showing that inactivation of BRCA2 triggers innate immune responses, which are potentiated by PARP inhibitors.

Overall the message of the manuscript is interesting and the results worth reporting.

I have the following comments:

1. The authors use two cell line systems to inducibly deplete BRCA2. I consider this to be a strength of the study, as it allows the authors to focus on gene changes that are shared in the two systems.
2. The authors observe an acute response consisting of downregulation of genes involved in cell cycle progression and a late response consisting of upregulation of interferon-inducible, innate immune response genes. The authors should discuss whether these two responses are related. According to the model of the authors, depletion of BRCA2 leads to DNA replication stress. The DRS could then both slow down cell cycle progression and activate the innate immune response.
3. Given that the authors observe changes in genes involved in cell cycle progression, the cell cycle data shown in Fig. 5c should be shown in Fig. 1. In fact, I wonder, if it would be better to show the data of Fig. 5c first, together with the data on expression of cell cycle regulated genes and then in a subsequent figure show the data of Fig. 5b, together with the upregulation of the innate immunity genes.
4. Panels Fig. 1c and 1d are not that informative. These panels should be moved to the supplementary section or not shown.
5. Panels Fig. 2e and 2f are also not very informative. They should be moved to the Supplementary section. The panel of Fig. 5b could be moved to Fig. 2.
6. Comparing Figs 3c and 5b, it seems that in one experiment IRF3 is phosphorylated at late time points after BRCA2 depletion and in the other from the early time points. Perhaps, there is a difference in these two experiments that I have missed.
7. The title of the manuscript places emphasis on PARP inhibitors. However, the experiments with the PARP inhibitors are not well developed. In Fig. 6a, only gene expression has been studied. Can the authors monitor cell cycle progression by FACS and IRF3 phosphorylation? Perhaps also H2Ax phosphorylation? Having the data in mice (Fig. 6b) is, of course, good, but some more characterization of the cells in tissue culture is warranted.
8. A recent study in Nature showed increased fork speed in cells treated with PARP inhibitors. The findings of the Nature study should be discussed. I am not asking that the authors repeat the findings reported in Nature in their system, but these findings should be discussed.

Overall, with some more experiments to augment Fig. 6a and with better organization of the way the findings are presented, this manuscript could become suitable for publication in Nature Communications.

Reviewer #2 (Remarks to the Author):

In this manuscript, the authors reported that short-term (4 days) or long-term (28 days) BRCA2 depletion exhibited a biphasic response in transcriptomic alterations analyzed by RNA-seq. Interestingly, long-term BRCA2 depletion induces cell –intrinsic immune signaling through activating of the cGAS-STING pathway. In contrast, short-term BRCA2 depletion causes cell cycle arrest in G1 and results in downregulation of genes involved in cell cycle progression, DNA replication and repair. Furthermore, the authors showed that treatment with PARP inhibitors stimulated the interferon response in cells and tumors lacking BRCA2, suggesting that therapeutic effect of PARPi on BRCA2-deficient cells may, in part, be mediated by activation of interferon signaling. This study presented an interesting observation, which is relevant to the role of BRCA2 deficiency in tumor biology. Results from this report may provide new insights into the mechanisms underlying the synthetic lethality interaction between BRCA2 deficiency and PARP inhibitors. However, the underlying mechanism how long-term BRCA2 loss, but not short-term BRCA2 loss, causes STING activation and activation of interferon signaling, is not clear. Several key issues need to be addressed.

1. The authors suggested that BRCA2 inactivation can lead to severe replication stress, aberrant chromosome segregation and persistent DNA damage, which conceivably may result in cytosolic DNA accumulation and activate interferon signaling. However no experimental data was provided to show whether there is an accumulation of cytosolic DNA in BRCA2-deficient cells in either short-term or long-term condition. It remains unknown whether short-term/ long-term BRCA2 loss may lead to differential effects on replication stress and chromosome segregation and whether this mechanism may explain the specific interferon signaling activation in long-term BRCA2 deficient cells.

2. In the long-term Dox activation experiments, an important control group using control shRNA targeting non-specific sequences or luciferase should be used. It will help exclude the possibility that long-term Dox activation of shRNA vector transcripts may activate interferon signaling, which may not be specific to BRCA2 loss. This is a major concern. Alternatively, reconstitution of BRCA2 expression using BRCA2 constructs resistant to the shRNA targeting sequences may help confirm the interferon signaling is activated due to BRCA2 loss.

3. A lot of cancer cell lines (breast or ovarian cancer cell lines) have known BRCA2 mutations, for example MDA-MB-436 cell line used in this manuscript, which have constitutive BRCA2 loss. Is interferon signaling activated compared to BRCA2-wildtype cell lines? A similar analysis could be easily conducted using cell line transcriptomic database as the TCGA data analysis in Figure 4b. Is this phenomena specific to BRCA2 loss or do BRCA1 mutant cells exhibit a similar phenomenon?

4. In Figure 5b, the time course analysis of Dox induction showed that the activation of STING expression, phosphor-IRF3 and phosphor-STAT1 did not exhibit a time course-dependent effects. BRCA2-depletion at short-term 4 days or 28 days led to expression of these proteins to a very similar degree. However the induction of interferon-related genes showed a time course-dependent effects. What could be the explanation to this discordance in signaling activation and interferon-related gene transcription? Data showed in this Figure 5b was not consistent with data showed in Figure 3c, where 4-days DOX induction did not activate a strong IRF3 or STAT1 phosphorylation.

5. In Figure S6a, STAT1 siRNA knockdown reduced expression of the innate immune response genes. However, compared to the control (without Dox), loss of BRCA2 still could induce a significant upregulation of these genes in STAT1 knockdown. It is not convincing to claim that BRCA2 inactivation-induced the innate immune response genes are STAT1-dependent. STING knockdown may be a better target to demonstrate the innate immune signaling is required. The effect of STAT1 knockdown is too broad, not specific to the innate immune signaling.

6. In Figure 6a, it is not clear whether Olaparib induces the innate immune gene expression in H1299 cells without Dox. It seems that all Olaparib-treatment data in this control group was normalized at "1". It is unclear whether Olaparib specifically-induced the innate immune gene activation in BRCA2-loss condition. In the xenograft tumors, only BRCA2-loss tumors were analyzed. BRCA2-reconstituted MDA-MB-436 should be a good control to support the conclusion.

7. In Figure S6, long-term loss of BRCA2 (about 40% inhibition) seemed to cause an increased resistance to Olaparib compared to short-term loss of BRCA2 (about 60% inhibition). This observation was not consistent with the conclusion that interferon signaling may contribute to

inhibitory effect of PARP inhibitors on BRCA2 loss cells since long-term BRCA2 loss led to a significant activation of the innate immune genes. It is also a concern that these BRCA2 knockdown cells were not very sensitive to PARPi.

Reviewer #3 (Remarks to the Author):

The paper by de Reisländer T. et al. reports the impact of BRCA2 depletion on cell transcriptional program. To do so, the authors employ a doxycycline-inducible shRNA expression system that depletes BRCA2 in H1299 and MDA-MB-231 cells. They find that the transcriptome of cells depleted of BRCA2 changes over time, with an early response characterized by reduced expression of cell proliferation genes whereas the transcriptomic response to long-term BRCA2 loss was characterized by cytokines and immune response genes. The authors then validate the findings related to induction of immune response genes in long-term BRCA2 depletion in cells and in a xenograft system. Interestingly they show that PARP inhibitor treatment induces induction of an innate immunity response in BRCA-deficient cells.

Overall, the manuscript is well written and this is an area of intense investigation in the field and the authors make a series of observations that will be of interest and shed new light into the mechanisms by which cells adapt to the loss of BRCA2-. While many of the conclusions are well supported by the experiments, some are on more shaky ground and will benefit from revision or qualification.

1. While the data seems solid, it relies on a technology, short hairpin-mediated RNA interference that is well known to induce innate immunity responses (e.g. see doi.org/10.1016/j.omtn.2016.12.007). The authors must provide control experiments where a an unrelated hairpin RNA is used as negative control. As it stands, it is unclear if the responses observed are due to a depletion of the BRCA2 protein or are rather due to a response induced by shRNA expression.
2. The model put forward by the authors suggests that the loss of BRCA2 leads to accumulation of cytosolic DNA that is sensed by the cGAS-STING pathway. Given the possibility that the response is due to shRNA expression (see point above), it would be important to firm up this hypothesis by (1) assessing the presence of cytosolic DNA and/or micronuclei and (2) testing whether depletion of STING or cGAS abrogates innate immune response to BRCA2 loss.
3. The data in figure 3b-c lead the authors to describe an activation of IRF3 and STAT1 after long-term depletion of BRCA2. However, the data in figure 5b seems to contradict these observations as IRF3 and STAT1 phosphorylation is observed in the earliest time points in the same cell line. How do the authors explain this discrepancy?
4. The data supporting that STAT1 inactivation rescues olaparib sensitivity in BRCA2-depleted cells (p11 and Sup Fig 6) is weak and is not controlled for non-targeting effects (as it uses only 1 siRNA). As it seems peripheral to the story, I suggest either to remove this data altogether or to firm it up with adequate siRNA controls such as multiple independent siRNAs or, even better, rescue the phenotype with an siRNA-resistant STAT1 transgene. A
5. MDA-MB-438 cells have a mutation in the BRCA1 gene rather than in BRCA2. The authors should be clear about this fact in their manuscript.
6. It is not clear why, in Fig 4, the two tumor groups being compared are BRCA2-high vs BRCA2-del. Why are tumors with high BRCA2 expression used a comparator on not tumor with median BRCA2 expression? Since mutations in other HR genes (BRCA1, PALB2 e.g.) are also found in HGSOC samples, would it not be more appropriate to compare HR-defective vs HR-proficient

tumors?

Minor Points

- On page 2, the authors indicate "[...] BRCA2 inactivation in tumors is associated with uncontrolled cell proliferation." Is this really the case? I guess in some ways, all cancers are associated with uncontrolled cell proliferation but the statement seems to point to a function for BRCA2 loss in the dysregulation of proliferation, which I doubt is the case. The authors should either revise the statement or provide a reference supporting this statement.

- On page13, second paragraph: "best." Seems to be a typo

We, the authors, are grateful to the Referees for their constructive comments on our manuscript. We have addressed all the points raised by the Referees, which significantly improved our manuscript.

Point-by-point response:

Referee #1:

This is an interesting manuscript showing that inactivation of BRCA2 triggers innate immune responses, which are potentiated by PARP inhibitors.

Overall the message of the manuscript is interesting and the results worth reporting.

I have the following comments:

1. The authors use two cell line systems to inducibly deplete BRCA2. I consider this to be a strength of the study, as it allows the authors to focus on gene changes that are shared in the two systems.

2. The authors observe an acute response consisting of downregulation of genes involved in cell cycle progression and a late response consisting of upregulation of interferon-inducible, innate immune response genes. The authors should discuss whether these two responses are related. According to the model of the authors, depletion of BRCA2 leads to DNA replication stress. The DRS could then both slow down cell cycle progression and activate the innate immune response.

Response: The replication stress characteristic of the acute BRCA2 inactivation could indeed slow down cell cycle progression and trigger, in the long term, the innate immune response, detected by upregulation of interferon-inducible genes. We have included a discussion of this important concept on p. 15 (middle paragraph) of the revised manuscript. Also, in Figure 2f we show new data indicating that that frequency of cGAS-positive micronuclei (which conceivably trigger the interferon response) increases from day 4 to day 28 of BRCA2 abrogation.

3. Given that the authors observe changes in genes involved in cell cycle progression, the cell cycle data shown in Fig. 5c should be shown in Fig. 1. In fact, I wonder, if it would be better to show the data of Fig. 5c first, together with the data on expression of cell cycle regulated genes and then in a subsequent figure show the data of Fig. 5b, together with the upregulation of the innate immunity genes.

4. Panels Fig. 1c and 1d are not that informative. These panels should be moved to the supplementary section or not shown.

5. Panels Fig. 2e and 2f are also not very informative. They should be moved to the Supplementary section. The panel of Fig. 5b could be moved to Fig. 2.

Response: In response to Reviewer's suggestion, we have rearranged the data in Figure 2: we moved the gene networks previously shown in Figures 2e and 2f to Supplementary section and included in Figure 2 the cell cycle profile and micronuclei activation triggered upon BRCA2 depletion. The new outline of Figure 2 clearly illustrates how cell cycle slowdown and micronuclei formation correlate with the early and late stages of BRCA2 inactivation. The text was correspondingly modified on p. 8 (top 2 paragraphs) of the revised manuscript. Panels in Figure 1c and 1d were kept because they illustrate how loss of BRCA2 alters the transcriptional profile from early to late time points.

6. Comparing Figs 3c and 5b, it seems that in one experiment IRF3 is phosphorylated at late time points after BRCA2 depletion and in the other from the early time points. Perhaps, there is a difference in these two experiments that I have missed.

Response: We have corrected this inconsistency in the revised version of the manuscript. The data in both Figure 3c and Figure 4b are now showing that IRF3 phosphorylation is detectable at both early and late timepoints.

7. The title of the manuscript places emphasis on PARP inhibitors. However, the experiments with the PARP inhibitors are not well developed. In Fig. 6a, only gene expression has been studied. Can the authors monitor cell cycle progression by FACS and IRF3 phosphorylation? Perhaps also H2Ax phosphorylation? Having the data in mice (Fig. 6b) is, of course, good, but some more characterization of the cells in tissue culture is warranted.

Response: We agree with the Reviewer's point that the effect of olaparib on the immune response was insufficiently developed. In the new version of our manuscript (new Figure 5) we included *in vitro* cell culture data to strengthen this part. We monitored cell cycle progression, induction of DNA damage (via KAP1 phosphorylation, a well-characterised ATM target) and IRF3 phosphorylation upon treatment with olaparib. In addition to interferon-stimulated gene (ISG) upregulation in cells treated with olaparib, we demonstrate in the revised manuscript that the same genes are upregulated specifically in BRCA1/2-deficient tumours treated with PARP inhibitor, using two xenograft models derived from MDA-MB-436 (*BRCA1*^{-/-}) and HCT116 (*BRCA2*^{-/-}) human cells (new Figure 6). Taken together, these results substantiate the concept that PARP inhibitors potentiated the innate immune responses in BRCA-deficient cells and tumours.

8. A recent study in Nature showed increased fork speed in cells treated with PARP inhibitors. The findings of the Nature study should be discussed. I am not asking that the authors repeat the findings reported in Nature in their system, but these findings should be discussed.

Response: That olaparib may increase fork speed is an interesting concept. Our own results showing that the frequency of S-phase (EdU-positive) BRCA2-deficient cells is reduced by olaparib treatment (new Figure 5e) seems consistent this concept. We refer to this publication and its relevance to our results in p. 13 (top paragraph) of our new manuscript.

Overall, with some more experiments to augment Fig. 6a and with better organization of the way the findings are presented, this manuscript could become suitable for publication in Nature Communications.

Referee #2:

In this manuscript, the authors reported that short-term (4 days) or long-term (28 days) BRCA2 depletion exhibited a biphasic response in transcriptomic alterations analyzed by RNA-seq. Interestingly, long-term BRCA2 depletion induces cell –intrinsic immune signaling through activating of the cGAS-STING pathway. In contrast, short-term BRCA2 depletion causes cell cycle arrest in G1 and results in downregulation of genes involved in cell cycle progression, DNA replication and repair. Furthermore, the authors showed that treatment with PARP inhibitors stimulated the interferon response in cells and tumors lacking BRCA2, suggesting that therapeutic effect of PARPi on BRCA2-deficient cells may, in part, be mediated by activation of interferon signaling. This study presented an interesting observation, which is relevant to the role of BRCA2 deficiency in tumor biology. Results from this report may provide new insights into the mechanisms underlying the synthetic lethality interaction between BRCA2 deficiency and PARP inhibitors. However, the underlying mechanism how long-term BRCA2 loss, but not short-term BRCA2 loss, causes STING activation and activation of interferon signaling, is not clear. Several key issues need to be addressed.

Response: We agree that the concerns expressed by this Reviewer are indeed critical. To address them, we included in our manuscript new data showing that micronuclei frequency increases with time upon BRCA2 inactivation (new Fig. 2f), which supports the mechanism of STING activation by cytosolic DNA accumulation and that interferon signaling is exclusively triggered by BRCA2 inactivation and not by shRNA-mediated long-term depletion of other factors (e.g. ERK1 and ERK2; Supplementary Fig. 8), highlighting the BRCA2 specificity of the observed response. The latter is indeed a key control experiment and our data unambiguously demonstrates now that BRCA2 inactivation itself triggers the immune response.

1. The authors suggested that BRCA2 inactivation can lead to severe replication stress, aberrant chromosome segregation and persistent DNA damage, which conceivably may result in cytosolic DNA accumulation and activate interferon signaling. However no experimental data was provided to show whether there is an accumulation of cytosolic DNA in BRCA2-deficient cells in either short-term or long-term condition. It remains unknown whether short-term/ long-term BRCA2 loss may lead to differential effects on replication stress and chromosome segregation and whether this mechanism may explain the specific interferon signaling activation in long-term BRCA2 deficient cells.

Response: To address whether cytosolic DNA accumulation activates interferon signaling, we quantified cGAS-positive micronuclei frequency in the short-term and long-term BRCA2 inactivation. Our results indicate that the levels of cGAS-positive micronuclei increase significantly from day 4 to day 28. These new data, which provide a link between the replication stress, DNA damage accumulation and interferon signalling in BRCA2-deficient cells are shown in the new Fig 2f.

2. In the long-term Dox activation experiments, an important control group using control shRNA targeting non-specific sequences or luciferase should be used. It will help exclude the possibility that long-term Dox activation of shRNA vector transcripts may activate interferon signaling, which may not be specific to BRCA2 loss. This is a major concern. Alternatively, reconstitution of BRCA2 expression using BRCA2 constructs resistant to the

shRNA targeting sequences may help confirm the interferon signaling is activated due to BRCA2 loss.

Response: To address the key point of whether interferon signaling is activated specifically by loss of BRCA2 and not by non-specific shRNA induction, we used two control cell lines expressing doxycycline (DOX)-inducible shRNAs against ERK1 or ERK2. Our results, included in the new Supplementary Fig. 8 and p. 10 (bottom paragraph), clearly demonstrate that upregulation of interferon-stimulated genes is specific to BRCA2 inactivation and does not occur in ERK1/2-depleted cells. Moreover, IRF3 and STAT1 phosphorylation are similarly detected only upon induction of BRCA2 shRNA.

3. A lot of cancer cell lines (breast or ovarian cancer cell lines) have known BRCA2 mutations, for example MDA-MB-436 cell line used in this manuscript, which have constitutive BRCA2 loss. Is interferon signaling activated compared to BRCA2-wildtype cell lines? A similar analysis could be easily conducted using cell line transcriptomic database as the TCGA data analysis in Figure 4b. Is this phenomena specific to BRCA2 loss or do BRCA1 mutant cells exhibit a similar phenomenon?

Response: MDA-MB-436 cells carry a 5396+1G>A mutation in BRCA1 which results in loss of BRCA1 protein expression (Elstrodt *et al*, *Cancer Res* 2006). We have clarified this in the revised version of our manuscript on p. 13 as well as in Fig. 6 (also requested by Reviewer 3, point 5). As a second BRCA2-mutated cells line, we used in our manuscript BRCA2-deleted DLD1 cells (Supplementary Fig. 7). When compared to the wild type controls, the transcriptome of BRCA2^{-/-} DLD1 cells shows a striking deregulation of interferon signalling. At the suggestion of this Reviewer (also expressed by Reviewer 3, point 6), we performed additional TCGA analyses similar to the one in previous Fig. 4b. We have included a comparison between BRCA2- or RAD51-deleted groups and a median control group in Figure 5a,b. Moreover, in the new Supplementary Fig. 10. we included a comparison between HR-compromised tumours (BRCA1/BRCA2/RAD51/PALB2-deleted) and HR-proficient control tumours which show upregulation of interferon-stimulated gene expression in HR-deficient tumours.

4. In Figure 5b, the time course analysis of Dox induction showed that the activation of STING expression, phosphor-IRF3 and phosphor-STAT1 did not exhibit a time course-dependent effects. BRCA2-depletion at short-term 4 days or 28 days led to expression of these proteins to a very similar degree. However the induction of interferon-related genes showed a time course-dependent effects. What could be the explanation to this discordance in signaling activation and interferon-related gene transcription? Data showed in this Figure 5b was not consistent with data showed in Figure 3c, where 4-days DOX induction did not activate a strong IRF3 or STAT1 phosphorylation.

Response: This discrepancy was also highlighted by Reviewer 1, point 6 and Reviewer 3, point 3. We addressed it in repeat experiments and found a consistent pattern of IRF3 and STAT1 phosphorylation at the early time points (day 4) in data shown in Fig. 3c and Fig. 4b. New immunoblots are included in the new Fig. 3c,d.

5. In Figure S6a, STAT1 siRNA knockdown reduced expression of the innate immune response genes. However, compared to the control (without Dox), loss of BRCA2 still could induce a significant upregulation of these genes in STAT1 knockdown. It is not convincing to claim that BRCA2 inactivation-induced the innate immune response genes are STAT1-dependent. STING knockdown may be a better target to demonstrate the innate immune

signaling is required. The effect of STAT1 knockdown is too broad, not specific to the innate immune signaling.

Response: We agree with the Reviewer's suggestion that the observed upregulation of the immune response genes may be specifically dependent on STING, rather than on STAT1, which is known to regulate a broad spectrum of genes. We addressed this point by depleting STING with siRNA (from day 1 to day 12 of DOX addition) and monitoring interferon regulated gene expression over time in both BRCA2-proficient and deficient cells. Our data, included in the new Supplementary Fig. 9a and p.11 of the revised manuscript, demonstrate that upregulation of these genes in BRCA2-deficient cells is indeed dependent on STING activity.

6. In Figure 6a, it is not clear whether Olaparib induces the innate immune gene expression in H1299 cells without Dox. It seems that all Olaparib-treatment data in this control group was normalized at "1". It is unclear whether Olaparib specifically-induced the innate immune gene activation in BRCA2-loss condition. In the xenograft tumors, only BRCA2-loss tumors were analyzed. BRCA2-reconsituted MDA-MB-436 should be a good control to support the conclusion.

Response: To address the Reviewer's question, we generated a new xenograft model using HCT116 BRCA2^{+/+} and BRCA2^{-/-} cells. Our data shown in the new Fig. 6c demonstrate that innate immune response genes are upregulated specifically in BRCA2-deficient tumours upon PARP inhibitor treatment. This is consistent with the response observed in the BRCA1-deficient MDA-MB-436 xenografts (Fig. 6b). A discussion of these results is included in p. 13 of the revised manuscript.

7. In Figure S6, long-term loss of BRCA2 (about 40% inhibition) seemed to cause an increased resistance to Olaparib compared to short-term loss of BRCA2 (about 60% inhibition). This observation was not consistent with the conclusion that interferon signaling may contribute to inhibitory effect of PARP inhibitors on BRCA2 loss cells since long-term BRCA2 loss led to a significant activation of the innate immune genes. It is also a concern that these BRCA2 knockdown cells were not very sensitive to PARPi.

Response: Following the concerns of this Reviewer and also the advice of Reviewer 3 (point 4) this part has been removed from the new manuscript.

Referee #3:

The paper by de Reisländer T. et al. reports the impact of BRCA2 depletion on cell transcriptional program. To do so, the authors employ a doxycycline-inducible shRNA expression system that depletes BRCA2 in H1299 and MDA-MB-231 cells. They find that the transcriptome of cells depleted of BRCA2 changes over time, with an early response characterized by reduced expression of cell proliferation genes whereas the transcriptomic response to long-term BRCA2 loss was characterized by cytokines and immune response genes. The authors then validate the findings related to induction of immune response genes in long-term BRCA2 depletion in cells and in a xenograft system. Interestingly they show that PARP inhibitor treatment induces induction of an innate immunity response in BRCA-deficient cells.

Overall, the manuscript is well written and this is an area of intense investigation in the field and the authors make a series of observations that will be of interest and shed new light into the mechanisms by which cells adapt to the loss of BRCA2-. While many of the conclusions are well supported by the experiments, some are on more shaky ground and will benefit from revision or qualification.

1. While the data seems solid, it relies on a technology, short hairpin-mediated RNA interference that is well known to induce innate immunity responses (e.g. see doi.org/10.1016/j.omtn.2016.12.007). The authors must provide control experiments where a an unrelated hairpin RNA is used as negative control. As it stands, it is unclear if the responses observed are due to a depletion of the BRCA2 protein or are rather due to a response induced by shRNA expression.

Response: We completely agree with the Reviewer's view and recognize that this is a key control for the specificity of the observed immune response to BRCA2 inactivation. The same concern was also raised by Reviewer 2 (Introduction and point 2). As outlined in our response to Reviewer 2, we monitored induction of interferon response genes in two other cell lines harbouring doxycycline-inducible shRNAs targeting either ERK1 or ERK2. In both of these cell lines, addition of doxycycline induced specifically abrogation of ERK1/2 but it did not induce a detectable interferon response (new Supplementary Fig. 8 and p. 10, bottom paragraph).

2. The model put forward by the authors suggests that the loss of BRCA2 leads to accumulation of cytosolic DNA that is sensed by the cGAS-STING pathway. Given the possibility that the response is due to shRNA expression (see point above), it would be important to firm up this hypothesis by (1) assessing the presence of cytosolic DNA and/or micronuclei and (2) testing whether depletion of STING or cGAS abrogates innate immune response to BRCA2 loss.

Response: We recognized that the experiments suggested by the Reviewer will strengthen our conclusion that cGAS-STING-STAT1 pathway activation mediates the innate immune response upon chronic BRCA2 abrogation. In our revised manuscript we provide new experimental evidence for increased cGAS-positive micronuclei in BRCA2-deficient cells from day 4 to day 28 of DOX addition (new Fig. 2f) and of attenuation of the innate immune response in STING-depleted cells (new Supplementary Fig. 9). These findings are also detailed in our response to Reviewer 2, point 1 and point 5.

3. The data in figure 3b-c lead the authors to describe an activation of IRF3 and STAT1 after long-term depletion of BRCA2. However, the data in figure 5b seems to contradict these observations as IRF3 and STAT1 phosphorylation is observed in the earliest time points in the same cell line. How do the authors explain this discrepancy?

Response: As indicated in our answer to Reviewer 1, point 6 and Reviewer 2, point 4, we addressed this discrepancy in subsequent experiments and found a consistent pattern between IRF3 and STAT1 phosphorylation at the early time points (day 4) in data shown in Fig. 3c and Fig. 4b. These new data were included in the new Fig. 3c,d.

4. The data supporting that STAT1 inactivation rescues olaparib sensitivity in BRCA2-depleted cells (p11 and Sup Fig 6) is weak and is not controlled for non-targeting effects (as it uses only 1 siRNA). As it seems peripheral to the story, I suggest either to remove this data altogether or to firm it up with adequate siRNA controls such as multiple independent siRNAs or, even better, rescue the phenotype with an siRNA-resistant STAT1 transgene.

Response: We agree with the Reviewer and have removed these results from the revised manuscript.

5. MDA-MB-438 cells have a mutation in the BRCA1 gene rather than in BRCA2. The authors should be clear about this fact in their manuscript.

Response: That is an important point (also raised by Reviewer 2, point 3) and we provide precise details of the *BRCA1* mutation of MDA-MB-436 cells in the revised manuscript (p. 13, middle paragraph and Fig. 6b). Moreover, we generated a second xenograft model using HCT116 *BRCA2*^{-/-} and *BRCA2*^{+/+} cells to strengthen the concept that PAPRi potentiates the immune response in BRCA2-deficient tumours. Upon treatment with the PARP inhibitor talazoparib we found elevated levels of interferon stimulated genes specifically in the BRCA2-deficient tumours (new Fig. 6c).

6. It is not clear why, in Fig 4, the two tumor groups being compared are BRCA2-high vs BRCA2-del. Why are tumors with high BRCA2 expression used a comparator on not tumor with median BRCA2 expression? Since mutations in other HR genes (*BRCA1*, *PALB2* e.g.) are also found in HGSOC samples, would it not be more appropriate to compare HR-defective vs HR-proficient tumors?

Response: BRCA2-high vs BRCA2-low were initially selected as a means to compare groups of similar size. In the revised version of our manuscript, we have included a comparison between tumours with median *BRCA2* expression and *BRCA2*-deleted tumours, as this Reviewer suggested. The data recapitulate the results shown in previous Fig. 4b. In addition, following the Reviewer's suggestion, we analysed expression of the innate response genes in tumours lacking other HR genes (i.e. *BRCA1*, *RAD51* or *PALB2*). The results obtained with tumours carrying these gene deletions have been pulled together for tumours carrying deletions in *BRCA1* ($n = 1$), *PALB2* ($n = 1$), *RAD51* ($n = 11$) and *BRCA2* ($n = 4$) genes.

Minor Points

- On page 2, the authors indicate “[...] BRCA2 inactivation in tumors is associated with uncontrolled cell proliferation.” Is this really the case? I guess in some ways, all cancers are associated with uncontrolled cell proliferation but the statement seems to point to a function for BRCA2 loss in the dysregulation of proliferation, which I doubt is the case. The authors should either revise the statement or provide a reference supporting this statement.

Response: We agree with the Reviewer that the Abstract text may have been misleading and changed it to: “Contrary to non-cancerous cells, where BRCA2 deletion causes cell cycle arrest or cell death, tumors carrying BRCA2 inactivation continue to proliferate.”

As an additional clarification: in our cellular models, we find that BRCA2 loss does elicit alterations in cell cycle profile (new Fig. 2e) and lowers the rate of cell proliferation (new Supplementary Fig. S4). So BRCA2 may control cell proliferation, although not directly.

- On page 13, second paragraph: “best.” Seems to be a typo

Response: We have made this correction and would like to thank the Reviewer for pointing it out.

REVIEWERS' COMMENTS:

Reviewer #1 (Remarks to the Author):

The authors have addressed my comments. Particularly, the experiment showing increased number of micronuclei after depletion of BRCA2, provides mechanistic insights for activation of the innate immune response at late time points.

I recommend that the manuscript be accepted for publication.

Reviewer #3 (Remarks to the Author):

I thank the authors for a thoughtfully revised manuscript.